**RESEARCH**  

# Removing reference bias and improving indel calling in ancient DNA data analysis by mapping to a sequence variation graph

Rui Martiniano[1†], Erik Garrison[2,3†], Eppie R. Jones[4], Andrea Manica[4] and Richard Durbin[1,2*]

*Correspondence:
richard.durbin@gen.cam.ac.uk
[†]Rui Martiniano and Erik Garrison contributed equally to this work.
[1]Department of Genetics, University of Cambridge, Cambridge, CB3 0DH UK
[2]Wellcome Sanger Institute, Cambridge, CB10 1SA UK
Full list of author information is available at the end of the article

## Abstract

**Background:** During the last decade, the analysis of ancient DNA (aDNA) sequence has become a powerful tool for the study of past human populations. However, the degraded nature of aDNA means that aDNA molecules are short and frequently mutated by post-mortem chemical modifications. These features decrease read mapping accuracy and increase reference bias, in which reads containing non-reference alleles are less likely to be mapped than those containing reference alleles. Alternative approaches have been developed to replace the linear reference with a variation graph which includes known alternative variants at each genetic locus. Here, we evaluate the use of variation graph software `vg` to avoid reference bias for aDNA and compare with existing methods.

**Results:** We use `vg` to align simulated and real aDNA samples to a variation graph containing 1000 Genome Project variants and compare with the same data aligned with `bwa` to the human linear reference genome. Using `vg` leads to a balanced allelic representation at polymorphic sites, effectively removing reference bias, and more sensitive variant detection in comparison with `bwa`, especially for insertions and deletions (indels). Alternative approaches that use relaxed `bwa` parameter settings or filter `bwa` alignments can also reduce bias but can have lower sensitivity than `vg`, particularly for indels.

**Conclusions:** Our findings demonstrate that aligning aDNA sequences to variation graphs effectively mitigates the impact of reference bias when analyzing aDNA, while retaining mapping sensitivity and allowing detection of variation, in particular indel variation, that was previously missed.

**Keywords:** Ancient DNA, Variation graph, Sequence alignment, Reference bias

## Background

In suitable conditions, DNA can survive for tens or even hundreds of thousands of years ex vivo, providing a unique window into the history of life [1]. Since the initial application of high-throughput sequencing to ancient human remains [2], the number of aDNA samples with available sequence data has been increasing at a fast pace, and currently, over 2000 ancient samples have been published [3]. These studies have provided insights into past population history and allow direct tests of hypotheses raised in archeology, anthropology, and linguistics [4, 5].

However, aDNA sequence analysis poses several significant challenges. The amount of DNA available is limited, and often only a small fraction is endogenous, coming from the target individual, with the rest originating from microbial contamination [6]. Read lengths are limited by the degradation of DNA due to taphonomic processes and subsequent environmental exposure. Post-mortem damage (PMD) of the DNA occurs at a high rate, introducing mismatches in DNA molecules, particularly in their tails which are frequently single-stranded or more exposed. This manifests mostly as the conversion of cytosine to uracil, but also can lead to depurination [1]. Ancient DNA may be treated with uracil-DNA-glycosylase (UDG) and endonuclease VIII to fully [7] or partially [8] remove uracil residues and abasic sites, leaving undamaged portions of the DNA fragments intact. However, this process results in a reduction of read length and library depth, which is disadvantageous. Furthermore, a number of unique and irreplaceable samples were sequenced prior to the adoption of UDG treatment. Taking all these factors into account, ancient DNA data is generally of low coverage, short length, and high intrinsic error rate.

The typical workflow for ancient DNA data processing starts with the alignment of sequencing reads to a linear reference genome, which contains only the reference allele at polymorphic sites. Reads containing the alternate allele are less likely to map than reads containing the reference allele, creating a potentially strong bias against non-reference variation, which can have a significant effect on population genetic inference and implications for many aDNA studies [9, 10]. For example, a standard approach to genotyping is to generate pseudo-haploid calls by selecting a random read crossing each variable site. However, because of reference bias, at heterozygous sites, reads containing the reference allele compose the majority of reads, resulting in a more frequent sampling of the reference allele than the alternate one.

There have been previous attempts to mitigate the effects of reference bias and low coverage in aDNA, such as by implementing a model of reference bias in genotyping [11], or by working with genotype likelihoods throughout all downstream population genetic analyses [12]. The use of different parameters with `bwa aln` can modulate the number of accepted mismatches to increase alignment sensitivity and in particular decreasing the `-n` edit distance parameter from the default value of 0.04 to 0.02 [13] or 0.01 [14] allows more mismatches and increases sensitivity. Recently proposed approaches modify the reference genome [10, 15] and/or the aDNA sequencing reads [10] in order to account for alternate alleles at polymorphic sites. The authors show that by taking into account non-reference variation in the alignment process, reference bias can be substantially reduced. However, a limitation of these approaches is that they have only considered biallelic single nucleotide polymorphisms (SNPs). Therefore, non-reference alleles at insertion and deletion (indel) loci are not accounted for, despite there being hundreds of thousands of non-reference

indels in a typical human genome [16], and these having a greater affect on read mapping than SNPs [17].

An alternative way to improve read mapping and avoid reference bias is to map reads to a sequence graph that represents both reference and alternate alleles at known variable sites [18]. However, the application of this approach to ancient DNA data has not yet been examined. We recently introduced the variation graph (vg) software [17], and in Fig. 1 show an example of how vg can recover the alignment of short aDNA reads to alternate alleles. Here, we apply vg and bwa aln systematically to map both simulated data and 34 previously published ancient human DNA samples, and demonstrate that mapping with vg can effectively reduce reference bias for ancient DNA samples, particularly for indels. Furthermore, vg increases sensitivity for detection of variation in aDNA, unlike read modification methods.

## Results

### Evaluating reference bias in aDNA using simulation

First, we used simulation to examine the impact of post-mortem deamination (PMD) in vg and bwa (aln and mem) read alignment, including assessments after applying sequencing read [10] and reference genome modification [15]. We generated all possible

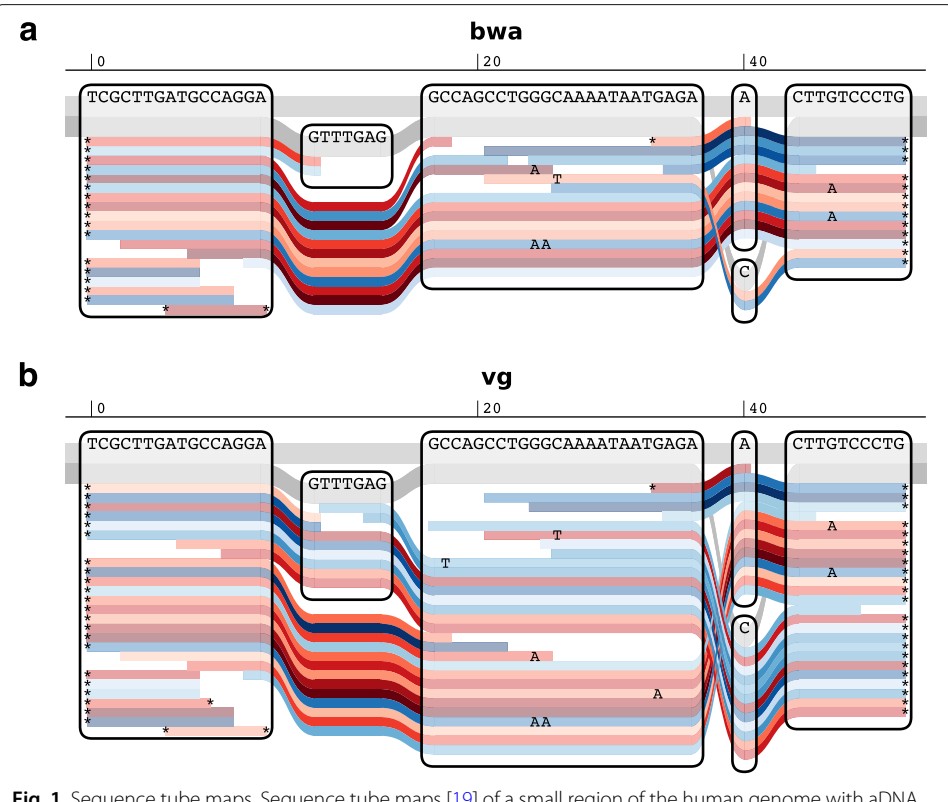

**Fig. 1** Sequence tube maps. Sequence tube maps [19] of a small region of the human genome with aDNA reads from the Yamnaya individual aligned with **a** bwa aln to a linear reference sequence and **b** vg map to a graph containing 1000 Genomes variants. The individual is heterozygous for both an indel (GTTTGAG/-) and a SNP (A/C) in this region, with insertion and alternate allele on the same haplotype. The two underlying haplotypes in this region are colored in gray, and red and blue lines indicate forward and reverse reads, respectively. None of the 6 reads across the insertion and only 2 of 12 reads across the SNP were mapped by bwa. Reads were locally realigned with vg map to the graph for the purpose of visualization

50-bp reads spanning variant sites on chromosome 11 of the Human Origins SNP panel [20, 21], which contains a set of SNPs designed to be highly informative about the genetic diversity in human populations. In half of the simulated reads, the SNP position was modified to carry the alternate allele. Different levels of ancient DNA PMD estimated in 102 ancient genomes from [22] were introduced into the reads using `gargammel` [23].

We generated a variation graph (1000GP graph) with variants identified as part of the phase 3 of the 1000 Genomes Project [16] above 0.1% minor allele frequency (MAF), to be used for read mapping with `vg`. We then mapped simulated reads back to the 1000GP graph or GRCh37 linear genome using `vg map` and `bwa aln`, respectively, and filtered the resulting alignments for those above mapping quality 30 for `bwa aln` aligned reads and mapping quality 50 for `vg` (see Additional file 1: Fig. S1 and Methods for details). The reason for using different mapping quality thresholds is that mapping qualities are estimated differently in `bwa aln` and `vg` and have different ranges: `bwa aln`'s maximum values are capped at 37 and `vg`'s at 60.

At high levels of simulated PMD, alignment with `bwa aln -n 0.02` against the linear reference prevents the observation of non-reference alleles in a large fraction of cases (Fig. 2a). This effect is notable at deamination rates as low as 10%, and with 30% deamination, the rate of alignment to non-reference alleles is reduced by nearly 15% relative to the total. In contrast, there is no such reduction for `vg map`. These observations are maintained across a range of different mapping quality thresholds (Additional file 1: Fig. S2). Given that we simulated the same number of reads at each SNP site, one with the reference allele and the other with the alternate, we would expect alternate and reference reads to be equally represented in the final alignments. However, because of reference bias, the fraction of alternate reads is on average 0.48267 95% CI [0.48095, 0.48438] in `bwa aln -n 0.02` but essentially 0.5 in `vg` 0.49988 95% CI [0.49984, 0.49991], supporting that `vg` alignment is not affected by reference bias in the same way as `bwa aln -n 0.02` (Additional file 1: Table S1).

When relaxing the edit distance parameter in `bwa aln` from -n 0.02 to -n 0.01 and increasing the maximum number of gap opens (-o 2), we observe as expected a higher sensitivity of mapping, and with it a better representation of alternate alleles 0.49702 95% CI [0.49657, 0.49747] in the final alignment, but the bias towards the reference is still slightly higher than with `vg` (Fig. 2b, d and Additional file 1: Fig. S2). Reducing the stringency in the mapping quality filter applied to the final alignments further improves the fraction of alternate reads mapped in both `bwa aln` (0.49936 95% CI [0.49927, 0.49945], -n 0.01 -o 2, mapQ $\geq$ 25) (Fig. 2b) and `vg graph` (0.50001 95% CI [0.50000, 0.50003], mapQ $\geq$ 30); however, as expected, decreasing mapping quality results in an increase in error rates (Additional file 1: Table S2).

In terms of sensitivity, using more permissive `bwa aln` parameters (-n 0.01 -o 2) mapped 99.60% (at mapQ $\geq$ 25) and 98.54% (at mapQ $\geq$ 30) of these reads, while `bwa aln -n 0.02` is less sensitive, resulting in only 98.56% (at mapQ $\geq$ 25) and 92.19% (at mapQ $\geq$ 30) mapped reads.

To discern whether these differences between `vg` and `bwa aln` are due to the use of a variation graph or the `vg` mapper, we also aligned simulated reads with the `vg` mapper to the human linear reference genome GRCh37 ("vg linear") and compared the results obtained with `vg` alignments to the 1000GP graph (Additional file 1: Fig. S3). In the `vg` alignment to the linear reference, the fraction of reads containing the reference allele that

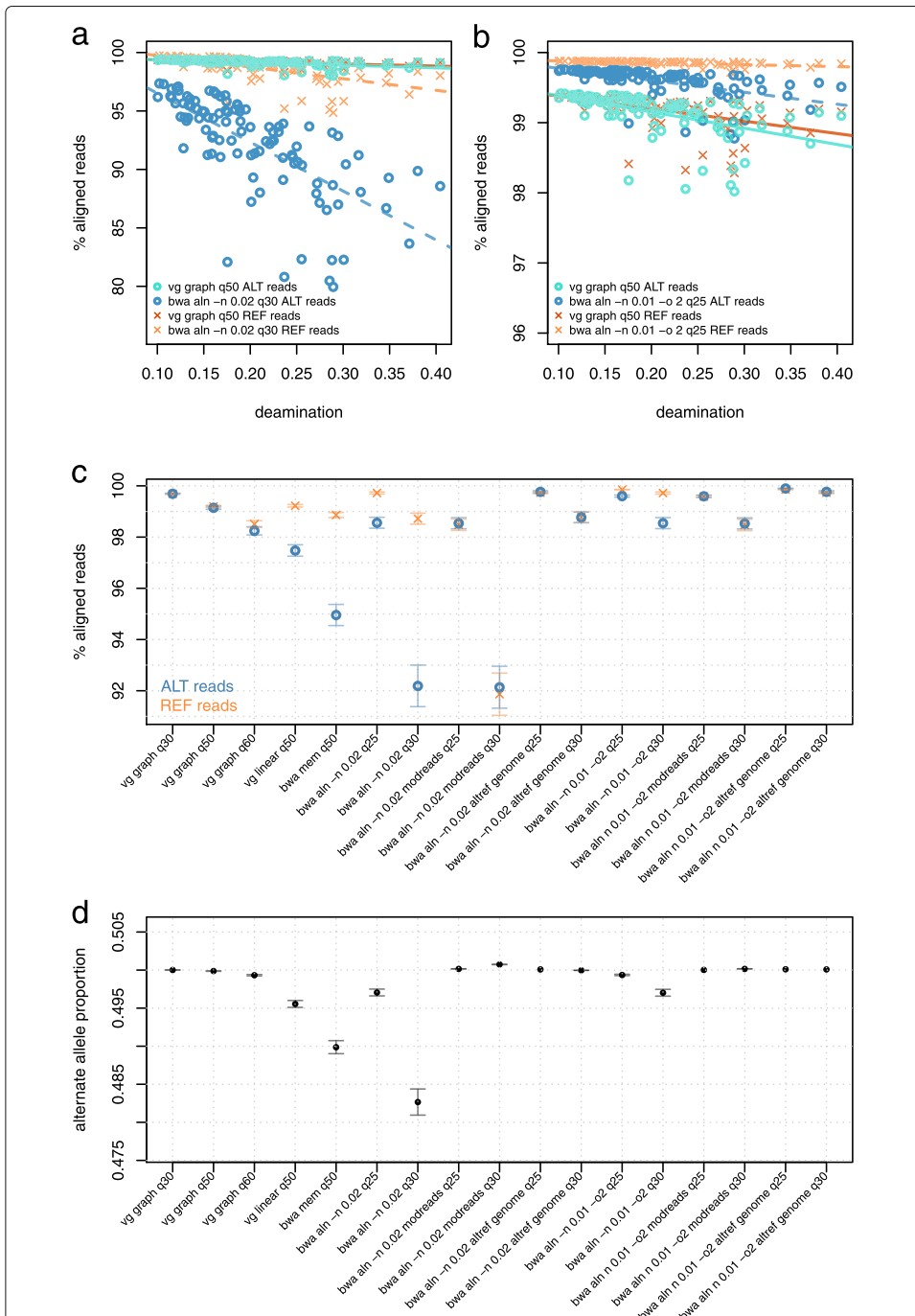

**Fig. 2** Comparing `vg graph`, `bwa aln`, and `bwa mem` using simulated ancient DNA. Comparing `bwa aln` and `vg map` performance when aligning reads simulated from chromosome 11 of the Human Origins panel. Lines represent ordinary least squares (OLS) regression results for the allele/aligner conditions corresponding to their colors. **a** Comparison between `vg graph` and `bwa aln -n 0.02`. **b** Comparison between `vg graph` and `bwa aln -n 0.01 -o 2`. **c** Comparison of the mean percentage (and 95% CI) of mapped reads in simulated data by `vg graph`, `bwa aln`, and `bwa mem` using different alignment parameters and minimum mapping quality filtering thresholds. **d** Mean alternate allele fraction (and 95% CI) of simulated reads after alignment with the different methods and minimum mapping quality filtering thresholds. We also show results obtained after processing simulated data with two previously published workflows for addressing reference bias: modified reads ("modreads") [10] and modified reference genome ("altref genome") [15]

are aligned remains constant at increasing rates of deamination, while, similarly to `bwa aln` and `bwa mem`, the percentage of aligned reads with the alternate allele drops as deamination increases.

We also applied the read modification protocol of Günther and Nettelblad [10] to our `bwa aln -n 0.02` mapping data, in which reads overlapping a biallelic SNP are duplicated with the copy carrying the other allele. If both reads map to the same region of the genome, then the mapping of the original, unmodified read is kept. In this case, the bias is removed (alternate allele fraction = 0.50074 95% CI [0.50071, 0.50077]), but at the cost of a substantial decrease in sensitivity for reads containing the reference (91.87%) as well as alternate alleles (92.14%) (Fig. 2c, d and Additional file 1: Fig. S4).

Applying the same workflow to less stringent `bwa aln` parameters (-n 0.01 -o 2, mapQ $\geq$ 25) greatly improves sensitivity (99.58% and 99.60%, for the reference and alternate allele, respectively) while effectively eliminating reference bias (alternate allele fraction = 0.50015 95% CI [0.50014, 0.50017]) (Additional file 1: Table S1).

We then processed our simulated data with a different workflow for removing reference bias as suggested by Peyrégne et al. [15]: reads are mapped to two versions of the human reference genome with `bwa aln`, one for each allelic version of the Human Origins SNPs. The resulting alignments are subsequently merged, keeping one random copy of the read if it maps to same genomic coordinates in both alignments and keeping also reads which map to one version of the reference genome, but not the other. This workflow was the most sensitive, mapping 99.90% (bwa aln -n0.01 -o2; mapQ $\geq$ 25; alternate allele fraction = 0.50011 95% CI [0.50011, 0.50011]) and 99.77% (at mapQ $\geq$ 30; alternate allele fraction = 0.50009 95% CI [0.50009, 0.50009]) of all alternate allele reads vs. 99.69% (at $q \geq$ 30) and 99.15% (at mapQ $\geq$ 50) with `vg graph` (Fig. 2c, d and Additional file 1: Fig. S5). However, despite its superior sensitivity, the Peyrégne et al. strategy comes at a cost of reduced accuracy in the mapping of reads containing the reference allele, as we demonstrate below.

We next examined the error rates of the various alignment strategies. We considered a given read to be correctly mapped if there was an exact match between the genomic coordinates from which it had been simulated and the ones for a major part of the alignment, taking into account any offsets introduced by insertions, deletions, and soft clips (soft clipping is the masking of a number of bases at the end of the read where they appear to be diverging significantly from the reference; this is done by read aligners to avoid misalignments around insertions and deletions, or problems with chimeric sequences).

In `vg graph` alignments, 1.2 per million reference allele reads and 2.5 per million alternate allele reads were incorrectly mapped. In `bwa aln` alignments, we observed that 0.2 and 2.4 per million reads containing the reference and the alternate allele, respectively, were incorrectly mapped. With more relaxed parameters (-n 0.01 -o 2), error fractions are slightly lower: 0.1 per million for reads carrying the reference allele and 2.0 per million for the ones with the alternate. With `vg` alignment to the linear reference sequence, reads containing the reference allele were mapped with similar accuracy to that observed in `vg` graph (1.3 per million), but the error in the alignment of reads with the alternate allele was one order of magnitude higher (11 per million) (Additional file 1: Table S2 and Fig. S6). The `bwa aln` read modification approach only removes excess reference allele reads, so it does not change the false positive rates for reads containing alternate alleles. The Peyrégne et al. [15] approach, however, requires the alignment of reads to an "alternate reference genome," which causes an increase in error rates, especially in the mapping

of reads containing the reference allele (11 per million at mapQ $\geq$ 25 and 5.3 per million at mapQ $\geq$ 30).

Error rates in all three of `vg graph`, `vg linear`, and `bwa aln` were positively correlated with the amount of deamination (Additional file 1: Fig. S7). There appears to be a qualitative difference between the types of errors made by `vg` and `bwa aln`, in that it makes more scattered errors, whereas `vg` tends to make clusters of errors at nearby locations (Additional file 1: Fig. S8). Unsurprisingly, the majority of errors ($\approx$ 70%) made by both methods occur in regions of reduced mappability (Additional file 1: Fig. S8).

To further investigate the false alignment rate of different read mappers, we aligned simulated microbial short reads (30–100 bp) with `vg` to the 1000GP graph and with `bwa aln` and `bwa mem` to the human reference genome (Additional file 1: Table S3 and Fig. S9). We observe distinct error patterns between the 3 aligners: in terms of short reads, `bwa aln -n 0.02` maps slightly more (0.897%, mapQ $\geq$ 30) microbial reads to the human genome than `vg` does to the graph (0.644%, mapQ $\geq$ 50), with the lowest percentage shown by `bwa mem` (0.001%, mapQ $\geq$ 50). Relaxing `bwa aln` parameters to "-n 0.01 -o 2" causes an increase (2.372%, mapQ $\geq$ 25) in the percentage of incorrectly mapped microbial reads compared to "-n 0.02" (Additional file 1: Table S3 and Fig. S9). When mapping longer fragments, both `vg graph` and `bwa mem` still present spurious alignments (0.111% and 0.234%, respectively, at read length of 70 bp), while with `bwa aln` with either value of `-n` virtually no microbial reads longer than 70 bp are aligned to the reference genome. As expected, the percentage of mapped microbial reads decreases when applying more stringent mapping quality filters to alignments generated by all three programs. Introducing different levels of deamination to microbial reads does not show a strong effect in their erroneous alignment to the human reference genome (Additional file 1: Fig. S10).

Together, the results of our analysis of simulated data demonstrate that the high degree of reference bias in ancient DNA read alignment when using `bwa` with standard parameters is mitigated at known sites by aligning against a variation graph with `vg`, or alternatively by relaxing the alignment parameters for `bwa aln`. Although read modification in `bwa aln` also removes bias, it does this at the cost of decreasing sensitivity for reads containing the reference allele, whereas `vg` increases the sensitivity for reads containing the alternate allele. This increase in `vg`'s sensitivity in mapping reads containing the alternate allele is achieved at comparable error rates to those observed with `bwa aln`, although there is a slight decrease in accuracy in mapping the reference allele.

### Aligning ancient samples to the 1000GP variation graph

To evaluate whether the results seen in the previous section carry over to real ancient DNA data, we selected 34 previously published ancient DNA samples (Table 1 and Additional file 1: Table S4), including Iron Age, Roman, and Anglo-Saxon period samples shotgun sequenced to low-medium coverage [24, 25], high-coverage Yamnaya and Botai culture individuals [26], and target captured samples from South America [27]. These are representative of the different types of data produced in the field of aDNA, as they are of variable genomic coverage, they were generated as part of SNP array target capture or whole-genome shotgun sequencing experiments and were subject to different enzymatic treatments.

**Table 1** Datasets analyzed in the present study

| Dataset | Number of individuals | Genomic coverage | Treatment | Type | Region |
|---|---|---|---|---|---|
| Damgaard et al. 2018 | 2 | 11.24-18.95x | Untreated | Whole-genome shotgun sequencing | Kazakhstan |
| Martiniano et al. 2016 | 9 | 0.54-1.63x | Untreated | Whole-genome shotgun sequencing | UK |
| Schiffels et al. 2016 | 10 | 0.47-7.86x | Partial UDG/USER | Whole-genome shotgun sequencing | UK |
| Posth et al. 2018 | 13 | 0.02-0.40x | Partial UDG | Target capture | South America |

First, we evaluated the effect of using `vg` on standard quality control metrics used in aDNA analysis, using mapping quality threshold 50 for `vg` and 30 for `bwa aln` as above, except where stated otherwise. When using ANGSD to estimate sample contamination from the X chromosome of male samples, `vg` gave similar but marginally increased values (mean 0.95%, range 0.30–3.36%) compared to `bwa aln -n 0.02` (0.87%, range 0.26–3.32%) (Additional file 1: Fig. S11). In terms of total endogenous DNA percentage, `vg` gave slightly lower percentages (Additional file 1: Table S5 and Additional file 1: Table S6), though as we will see below, more reads are mapped to alternate alleles. Finally, reads mapped with `vg` continue to show terminal deamination damage, which is used as a standard diagnostic for the presence of true ancient DNA, as seen in mapDamage [28] plots, although levels are slightly reduced (Additional file 1: Fig. S12). We attribute this reduction to differences in softclipping by the `vg` algorithm, which follows `bwa mem` not `bwa aln` (Additional file 1: Fig. S13).

To investigate the effect of using `vg` and a variation graph on genetic variant calling and genotyping, we focused on the Yamnaya sample from reference [26], which provides approximately 20-fold coverage of the genome, thus allowing us to compare results to confident genotype calls and to downsample to explore behavior at different sequencing depths. We called variants on the full depth sample using bcftools [29] for both `vg` and `bwa aln` alignments and used these callsets as ground truth. Looking at high-quality heterozygous transversion sites, `vg` has an alternate allele mapped read fraction of 0.4925 95% CI [0.4914,0.4937] compared to 0.4742 95% CI [0.4731, 0.4754] of `bwa aln -n 0.02` and to 0.4773 95% CI [0.4761, 0.4784] of `bwa aln -n 0.01 -o 2`. As expected, this difference was entirely due to mapping to previously identified 1000 Genomes Project sites present in the graph: new sites not in the graph showed no difference between the methods (Additional file 1: Fig. S14 and Fig. S15). The restriction to transversions for this analysis is a standard approach in aDNA analysis to control for noise created by deamination damage, which generates apparent transitions.

We next measured our ability to recover the heterozygous variants in the full coverage set at lower coverage levels. As seen in Fig. 3a, when calling using `bcftools`, `bwa aln` recovers fewer heterozygous SNPs than `vg map` alignment to the 1000GP graph at all coverage levels, regardless of the parameters used ("-n 0.02" or "-n 0.01 -o 2"). For example, at 4x coverage, `vg map` recovers ≈ 13% more heterozygotes as a fraction of the total. Filtering `bwa aln` alignments using read modification reduces sensitivity still further. Additionally, relaxing the mapping quality filter from 30 to 25 gave only a

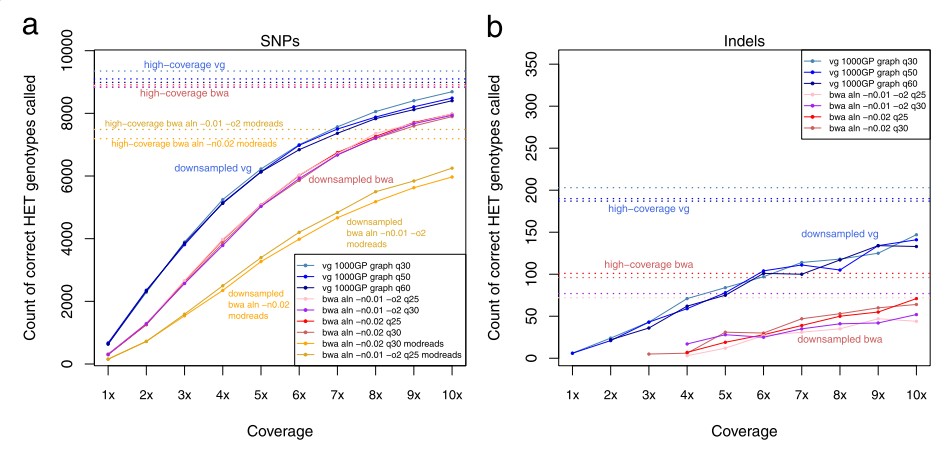

**Fig. 3** Downsampling a high-coverage aDNA sample. The comparative effect of downsampling on heterozygous variant calling following `bwa aln` and `vg map` alignment of reads from the ancient Yamnaya sample [26] with different parameters and mapping quality filtering thresholds, and including post-processing of `bwa aln` with the modified read filter [10]. **a** SNPs. **b** Indels (the modified read filter does not apply in this case)

marginally higher sensitivity to `bwa aln`. We note that if pseudo-haploid calls were made by selecting a random spanning read as is often done in aDNA analysis [30], then the allele imbalance described in the previous paragraph will directly lead to undercalling of alternate alleles.

The effect of reference bias in indel detection is even more striking. In Fig. 3b, `vg graph` recovers many more indels than `bwa aln`, approximately twice as many at high coverage and an even greater factor at lower coverage. If reference bias for indels were unrelated to allele length, then the average coverage of an alternate allele would be approximately constant across allele lengths. This is what we see with `vg graph` but not with `bwa aln`, which was unable to detect variants with allele length above 7 bp (Fig. 4a and Additional file 1: Fig. S16). This means that because of reference bias, we are missing important variation with `bwa aln` which is recoverable with `vg map`.

To illustrate this point, we looked at a clinically important variant associated with HIV-1 resistance (CCR5 delta 32), whose origins and history have been debated in the literature [31, 32]. This deletion was not detected in any of the ancient samples using "-n 0.02" or "-n 0.01 -o 2" `bwa aln` parameters, but was clearly present in the 4900-year-old Yamnaya sample and three more recent ancient British samples (Fig. 4b). The Yamnaya observation predates the previous oldest direct measurement in ancient skeletons 2900 years old [33], consistent with older dates of origin of the allele suggested by population genetic analysis [32]. The ability to detect the variant without bias enables investigation of the allele frequency trajectory in ancient samples.

### Population genetics analyses

In order to evaluate the consequences of reference bias, we applied the ABBA-BABA test of phylogenetic tree topology based on the $D$-statistic of population relationship [20, 34]. When estimating $D$-statistics of the form $D$(vg graph, bwa -n 0.02; GRCh37, Chimp), a deviation from zero indicates an excess of shared alleles between `bwa`- or `vg`-aligned samples and the GRCh37 reference genome. Our results based on pseudo-haploid

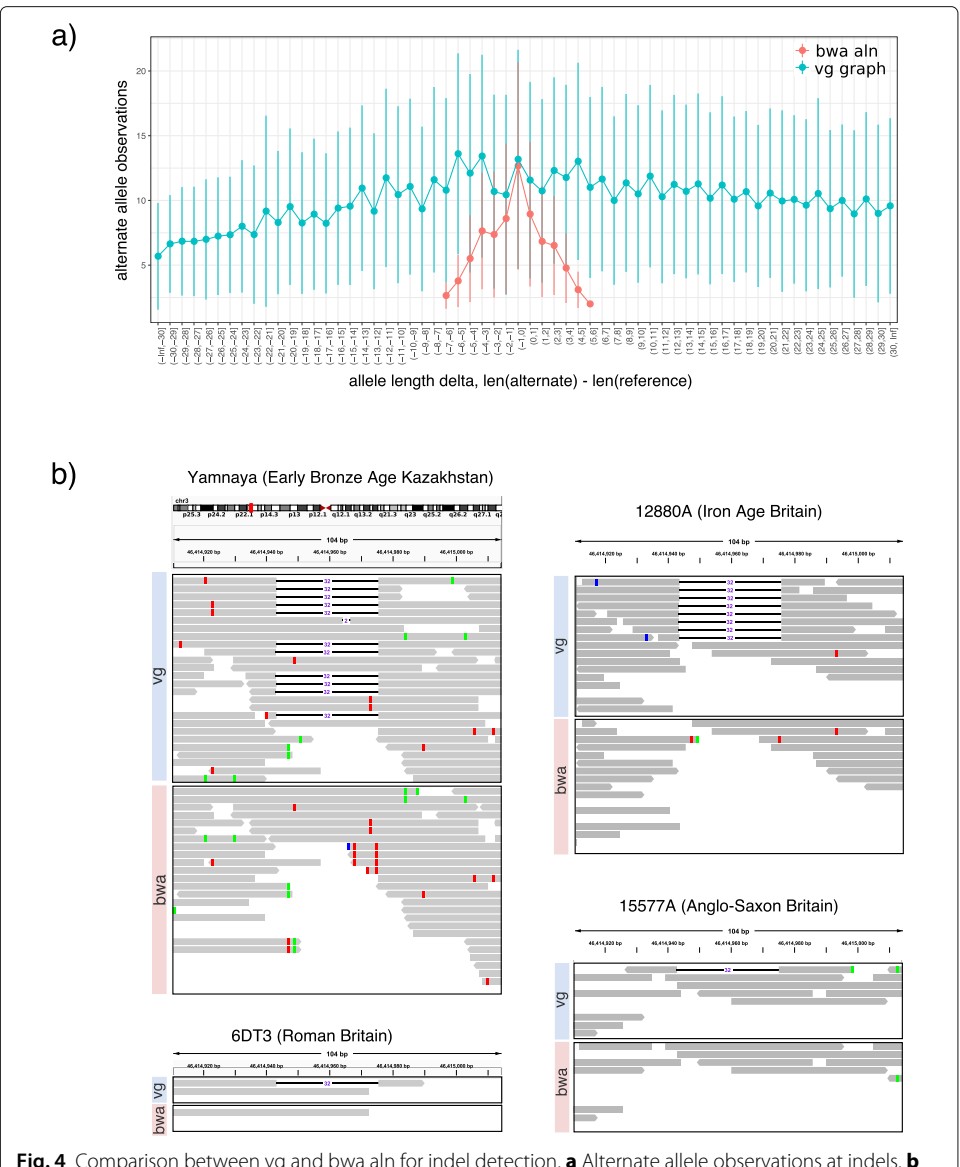

**Fig. 4** Comparison between vg and bwa aln for indel detection. **a** Alternate allele observations at indels. **b** Comparison between `vg graph` and `bwa aln` in the detection of the CCR5 delta 32 deletion associated with HIV-1 resistance. Reads containing the deletion were mapped with vg in four ancient samples, but not with bwa

random-allele calls, summarized in Additional file 1: Fig. S17, show negative *D*-statistics for all but a handful of samples when we use the `-n 0.02` settings, consistent with `bwa` calls being closer to the reference than `vg` calls (also observed with *D*(vg graph, bwa -n 0.02; GRCh37, Alternate Allele) (Additional file 1: Fig. S18)), but this bias is removed when the `bwa aln -n 0.01 -o 2` settings are used (Additional file 1: Fig. S17). We observe the same process in *D*-statistics with simulated data (Additional file 1: Fig. S19). When we applied the read modification approach to the `bwa`-mapped data, we also saw no consistent deviation from zero in *D*(vg, bwa-modreads; GRCh37, Chimp), as expected from our earlier results (Additional file 1: Fig. S20).

We also investigated the effect of `vg` or `bwa aln` alignment on Principal Component Analysis (PCA), another widely used analysis technique in the field of aDNA. Restricting

this analysis to samples from Europe and West/Central Asia, we projected the ancient samples and the reference genome onto a PCA plot derived from modern samples. We observe modest differences between the positions of `vg` and `bwa` aligned samples in the first two principal components, but these are not conclusive in terms of the direction of the bias (Additional file 1: Fig. S21 and Fig. S22). For example, the `bwa` processed Botai sample appears to be slightly closer to the reference than its `vg` aligned equivalent, while the opposite pattern is observed for the Yamnaya sample. Given the variability in our PCA results, it is not possible to make strong conclusions about the effects of removing reference bias on PCA projection.

Given the strong differences in terms of indel detection observed between `vg` and `bwa aln` processed data, we also investigated the impact of reference bias on PCAs estimated with indels of different lengths. When restricting our analysis to chromosome 21 alternate alleles called in the `vg` processed Yamnaya individual, clear genetic clusters corresponding to the 1000 Genomes super populations are maintained across all allelic lengths up to 18 bp (Additional file 1: Fig. S23). The same is not true for `bwa aln`, which did not recover any indels longer than 7 bp. This confirms that because of reference bias, when using standard methods for ancient DNA sequence alignment, population genetic analysis cannot reliably make use of indel data, although there is information present which can be accessed without bias when mapping with `vg`.

## Discussion

The analysis of highly fragmented and damaged ancient DNA sequence data is challenging and subject to reference bias, leading to a relative under-representation of alternate alleles at polymorphic sites. The consequences of this in downstream analysis can be real but quite subtle, as has been noted before [10], and we have seen in our results. Here, we have shown that `vg` can be used to effectively remove this reference bias, especially in the presence of post-mortem damage. In particular, it makes available unbiased calling of indel polymorphisms, which are frequently ignored in ancient DNA studies due to very strong reference bias.

Although other methods have recently been introduced to address reference bias in SNPs, all of these make some compromises. The approach to modify reads [10] reduces sensitivity, while the alternative to modify the reference [15] increases error rates. Our results suggest that the best approach to using `bwa aln` for ancient DNA alignment is probably to use very relaxed parameters, as proposed by Kircher et al. [14] (`-n 0.01 -o 2`) in combination with disabling seeding as recommended by [13] (`-l 1024`) and mapQ 25 filtering. This had a beneficial impact in terms of sensitivity (99.60% vs 97.34% for the mapping of reads containing the alternate allele) and a more balanced alternate allele representation (0.49936) but also increases error rates, for example increasing the rate of false mapping short microbial reads (Additional file 1: Fig. S9).

We have shown that `vg` effectively removes reference bias at known variants in its graph (both SNPs and indels), and its spurious alignment of microbial contaminants at short lengths can be controlled more effectively than for `bwa aln` with relaxed parameters by increasing the mapping quality threshold. Erroneous mapping of short contaminant sequences is a known issue in ancient DNA, and strategies are continuously being developed to address it [35]. The `vg` approach also uses alignment information efficiently for variant calling, which can be important at low read coverage (Fig. 3).

One complication of our analysis is that mapping qualities are not directly comparable between `vg` and `bwa aln`. Because of this, we presented comparisons between `vg` and `bwa aln` at different mapping quality filter thresholds. For `vg` in particular, we recommend imposing a minimum mapping quality filter of 50 for obtaining error rates comparable to those of `bwa aln` (albeit slightly higher), while maintaining high sensitivity and minimizing the spurious alignment of microbial reads.

More generally, we note that different mapping programs and parameters, and different procedures for data pre-processing such as adapter trimming, or the imposition of a minimum read length threshold prior alignment, will all affect how ancient samples compare to each other. For any given analysis, it is important to standardize these settings and to remap all ancient samples in the same way to reduce spurious findings.

An additional drawback of `vg` is the slightly lower sensitivity when compared to `bwa aln` in the mapping of reads in repetitive regions, as shown in [17]. When aligned to the linear reference, they map to a unique place in the linear reference, but in variation graphs they may map to more than one place. This becomes worse as more variants are introduced into the graph, which is why we included only variants with 0.1% minor allele frequency or more in our graph, as recommended by Garrison et al. [17].

Furthermore, read alignment with `vg` takes approximately 5× longer than with `bwa aln -n 0.01 -o 2` and 29× than `bwa mem` (Additional file 1: Table S7), and the memory requirements for both indexing the graph and read mapping can be much more substantial than for indexing a linear reference genome, depending on the number of variants included.

One possible concern with the use of `vg` as proposed is that it depends on a reference graph constructed from present-day human variation. For modern human samples from the last 50,000 years, this is not a major issue, since almost all common variation is shared with extant populations on that time frame. For example, 96.99% of high-quality variants called de novo in Ust'Ishim chromosome 1 accessible regions are found in the 1000 Genomes Project variant set [36]. However, this approach would not be appropriate for samples from archaic populations such as Neanderthals and Denisovans, for which we do not yet have substantial collections of genetic variation. Introgressed material from archaic humans within modern humans can provide a partial source of information on genetic variation in those parts of the genome where it persists, but for graph alignment approaches to work effectively across the whole genome in archaic samples, we will have to wait until sufficient archaic genomes have been sequenced to high depth to enable construction of a representative archaic variation graph. A related advantage of working with graph genomes is that, as multiple independently assembled human genomes (modern or ancient) are added into the reference variation graph, we will be able to assign ancient DNA sequence to human sequences not in the current reference graph, which are currently hidden from standard analyses of ancient DNA.

Beyond studies of human genetic history, ancient DNA is also increasingly used to study the history and evolution of other species, from bacterial pathogens to domesticated crops and extinct megafauna [3]. In many of these cases, natural diversity is higher than in humans, and "pangenomic" approaches that are equivalent to sequence variation graphs are becoming more widely used, often including more complex structural variation [37, 38]. Ancient DNA analyses in such species and systems are expected to benefit from a

variation graph mapping approach proportionately to the increased diversity represented in the pangenome variation graph.

Finally, as shown in our analyses, indel variants have the potential to be ancestry informative, but have been almost totally ignored in the aDNA field because of difficulties in aligning reads containing these variants, particularly when above a few base pairs length. Variation graph approaches offer a way of accessing this variation and open new avenues for aDNA research both at the level of population history but also by enabling probing of clinically relevant indel mutations in ancient individuals across the archeological record, as demonstrated for the CCR5 deletion allele.

## Methods

### Datasets and sequence data processing

In order to compare read mapping between `vg` and `bwa aln`, we compiled a dataset of sequencing reads from previously published ancient individuals (Table 1). Adapter trimming was done with AdapterRemoval [39] for paired reads (merging overlapping reads) and cutadapt [40] for single ended reads. Unaligned FASTQ data from the other two datasets [25, 27] were already provided with trimmed adapters. We aligned trimmed reads to the human linear reference genome (hs37d5) using `bwa aln` [29] with parameters -l1024 (for disabling seeding) and -n 0.02 [13] or -n 0.01 -o 2, with minimum base quality -q 15. We constructed the index file for `vg` [17] with hs37d5 and variants from the 1000 Genomes Project phase 3 dataset [16] above 0.1% MAF. In total, the graph contained 27,485,419 SNPs, 2,662,263 indels, and 4,753 other small complex variants. Trimmed reads were aligned to the variation graph using `vg` (v1.16.0-137-ge544284) map with parameters "–surject-to bam -k 15 -w 1024." Duplicate reads were removed with sambamba markdup [41] using the "–remove-duplicates" parameter. BAM files were subsequently filtered with samtools view [29], selecting reads with different mapping qualities thresholds (`bwa aln` and `vg`: mapQ > 0; $\geq 25$, $\geq 30$; `vg` only: $\geq 50$; $\geq 60$). The reason for using different mapping quality thresholds is that `bwa` uses a different mapping quality estimation process with maximum around 37 than `vg` with maximum 60. Coverage was estimated with qualimap [42] bamqc utility. We present read number, endogenous DNA content, and coverage for samples aligned with `vg` and `bwa aln` in Additional file 1: Table S5 and Table S6.

### Simulations

We simulated all possible reads overlapping chromosome 11 SNPs in the Human Origins dataset [20, 21]. In half of the simulated reads, the alternate allele was introduced. We then added different levels of deamination into simulated reads using `gargammel` [23], based on empirically estimated post-mortem damage in a dataset of 102 ancient genomes [22]. We aligned these simulated reads to the 1000GP graph with the `vg` mapper or to the linear human reference genome (GRCh37) with `bwa aln`, with parameters -n 0.02 or -n 0.01 -o 2 [14], `bwa mem` and `vg` (here referred to as "vg linear"). Read mapping with `vg` to the 1000GP graph took approximately four (2.12–7.57) times longer than with `bwa aln -n 0.02`. The resulting alignments were sorted with sambamba sort, converted to bam with samtools view, and filtered with different mapping qualities thresholds. We estimated read alignment accuracy by comparing the genome coordinate from where each read originates and the coordinate obtained after mapping, accounting for offsets between

these caused by softclips, deletions, and insertions. Read mapping errors were visualized using the R [43] package circlize [44]. To investigate the impact of different read lengths and deamination in the false alignment rates of the three read mappers (vg, bwa aln and bwa mem), we simulated 100,000 reads of different sizes (35–100 bp) from a set of microbial reference genomes identified in the Clovis sequence data [45] using gargammel [23]. Additionally, we introduced post-mortem changes in a subset of this data (30, 50, 70, and 90 bp) based on [22]. We processed all simulated microbial read data as described above.

### Authenticity and contamination estimates
Post-mortem deamination plots were generated with mapDamage v2 [28], sampling one million reads per sample. We estimated X-chromosome contamination in all male samples with ANGSD [46], with the parameters "-r X:5000000-154900000 -doCounts 1 -iCounts 1 -minQ 20" and using polymorphic sites identified in the HapMap Project.

### Variant calling and population genetics analyses
For population genetics analyses, we used the Human Origins dataset distributed with Lazaridis et al. [47]. In order to estimate D-statistics and Principal Component Analyses, we generated pileups for each individual [48] at 1233553 SNPs from the Human Origins dataset using samtools mpileup, disabling base quality score recalibration and imposing a minimum base quality filter of q20. We note that pileups were generated from bam files filtered with a minimum mapping quality threshold of 30 for bwa aln or 50 for vg. We generated pseudo-haploid genotypes by randomly sampling one allele at each SNP site and converted resulting pseudo-haploid genotypes to PLINK format using PLINK 1.9 [49]. These were subsequently merged with the Chimp and Href (the human reference genome) samples from the Human Origins dataset and converted to eigenstrat format using convertf. We estimated D-statistics with qpDstat [20], passing the parameter "printsd: YES" to obtain standard deviation estimates.

For the Principal Component Analysis estimated with SNP sites, we first filtered the Human Origins dataset, removing variants with minor allele frequency below 0.02 and genotyping missingness of 0.05, and selecting West Eurasian individuals. We merged this dataset with the pseudo-haploid genotypes belonging to the ancient samples as described above and ran smartpca [50, 51], restricting the analysis to transversion SNPs, using the parameters "lsqproject: YES" to project ancient samples into the PCA coordinates estimated with present-day populations, "killr2: YES" to exclude SNPs in high linkage disequilibrium (r2thresh: 0.2) and performing two iterations for outlier removal (numoutlieriter: 2).

We used PLINK to estimate PCAs with indels. We prepared our datasets by first calling indels in the Yamnaya sample processed with vg and bwa, as described below, keeping variants with quality equal or greater than 30 and keeping biallelic indels only. We used vt [52] for variant normalization, taking the human reference genome as input, and duplicate removal. Then, we generated two datasets, based on the 1000 Genomes chromosome 21 indels, restricting by variants with alternate alleles identified in the vg- or in the bwa-aligned Yamnaya sample.

### Downsampling experiment

We downsampled `bwa aln` and `vg` alignments belonging to the high-coverage Yamnaya individual from 1 to 10x using samtools. We then called 1,054,447 biallelic SNPs present in the 1000 Genomes chr21 VCF from all alignments using bcftools v. 1.8, requiring a base quality of at least 20. From the resulting variant calls, we kept only biallelic SNPs and selected heterozygous genotypes. We removed potential deamination SNPs and excluded variant calls with quality score below 30. Finally, we estimated the proportion of variants correctly recovered by comparing the genotypes obtained from the downsampled alignments with those obtained at full coverage. Comparison with the read modification method was done by modifying the downsampled and full coverage `bwa`-aligned reads with the 1000 Genomes SNP alleles and calling variants as described above.

### Alternate allele support and allele balance

In order to compare alternate allele support between `vg` and `bwa aln` alignments, we called chromosome 1 SNPs from the Yamnaya alignments with bcftools. We then filtered these by variant quality greater or equal than 30, with depth of coverage above 8, and selected heterozygous variants. From these genotype calls, we obtained reference and alternate allelic depth and compared alternate allele support between the `vg` and `bwa` aligned sample. To investigate reference bias at the level of indels, we called variants with FreeBayes [53] from the Yamnaya sample processed with both `vg` and `bwa aln` with default parameters, which we subsequently filtered for the sites present in the 1000 Genomes variation graph used for alignment.

### Comparison with additional methods for reducing reference bias

We compared `vg` with the workflow proposed by [10] to reduce reference bias. The following method was applied to both real and simulated data. First, for each `bwa`-aligned sample, we selected reads overlapping with the Human Origins SNPs or with the 1000 Genomes dataset. We then modified the allele in these reads using the "modify_read_alternative.py" script, distributed with [10], and remapped them with `bwa aln` to GRCh37 as described above. We then kept the original reads which mapped to the same location of the modified reads with "filter_sam_startpos_dict.py." We estimated *D*-statistics from the resulting filtered alignments as described above.

We also compared `vg` with a second workflow for removing reference bias [15]. Simulated sequence reads were aligned with bwa aln to two versions of the reference genome, one containing reference alleles and the other alternate alleles. We used "bam-mergeRef" (https://github.com/StephanePeyregne/bam-mergeRef) to merge the resulting alignments, keeping one version of a read if it maps to the same region in both alignments, and also keeping reads mapped in one alignment but not in the other.

### Supplementary information

---

**Additional file 1:** Figures S1-S23 and Tables S1-S7.

**Additional file 2:** Review history.

---

**Peer review information**

**Acknowledgements**
We thank Cosimo Posth and Stephan Schiffels for assistance in obtaining raw sequence data, Christiana Lyn Scheib and Toomas Kivisild for sharing their results on reference bias in ancient DNA and encouraging this study, Lara Cassidy for helpful discussions, and Torsten Günther and Gabriel Renaud for technical assistance.

**Review history**
The review history is available as Additional file 2.

**Authors' contributions**
RD conceived the work; all authors designed the experiments, analyzed, and interpreted the data; RM, EG, and RD wrote the manuscript. All authors read and approved the final manuscript.

**Funding**
RM was supported by an EMBO Long-Term Fellowship (No. ALTF 133-2017), EG by a Wellcome Sanger Institute studentship under grant WT206194, and RD by Wellcome grant WT207492. ERJ was supported by a Herchel Smith Fellowship and AM by the ERC Consolidator Grant 647787 (LocalAdaptation).

**Availability of data and materials**
Simulated raw sequence data generated in this study can be downloaded at Zenodo (DOI: 10.5281/zenodo.3416364) [54]. Raw sequence data for the samples analyzed in this study can be found in the European Nucleotide Archive, accessions PRJEB25389 [26], PRJEB11004 [24], ERP003900, and ERP006581 [25]. FASTQ files belonging to the South American individuals were provided by the authors of [27].

**Ethics approval and consent to participate**
All the sequencing read data analyzed in this work was previously published.

**Consent for publication**
Not applicable.

**Competing interests**
The authors declare that they have no competing interests.

**Author details**
[1]Department of Genetics, University of Cambridge,  Cambridge, CB3 0DH UK. [2]Wellcome Sanger Institute,  Cambridge, CB10 1SA UK. [3]Genomics Institute, University of California, Santa Cruz, CA 95064 USA. [4]Department of Zoology, University of Cambridge, Cambridge, CB2 3EJ UK.

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

## 
