## [**Additional file 2** Review history. · Genome Biology]

Review History

First round of review

Reviewer 1

Are you able to assess all statistics in the manuscript, including the appropriateness of statistical tests used? Yes, and I have assessed the statistics in my report.

Comments to author:

Martiniano et al. run the variance graph aligner vg on simulated and real ancient DNA datasets. As expected, the alignments show more equal representation of aligned reads supporting reference and (known) alternative alleles. Compared to one other method, that aims at alleviating reference bias in bwa alignments, they find an increased sensitivity.

While I have no doubt that vg is the right approach and can be useful for ancient DNA analyses, I have a number of issues with the methodology presented in the paper:

The authors use a mapping quality cutoff of 30 and parameters "-n 0.02 -l 1024" for bwa. More sensitive parameters (-n 0.01 instead of -n 0.02) have been used in the literature. It is also known that bwa sets mapping quality to 25 when reads align uniquely with the maximum edit distance. To reduce reference bias, a cutoff of MQ25 is often used. Fig. 2 shows quite clearly that the chosen BWA parameters perform poorly for both reference and alternative allele containing reads. It seems that more sensitive parameters, at least to the degree published in the literature, would be a fairer comparison.

The greater sensitivity compared with Günther & Nettleblad (2019) is likely explained by the choice of bwa parameters (see above) and the fact that Günther & Nettleblad use a third allele when modifying the reference. In a different approach, Peyrégne et al. (2019) (DOI: 10.1126/sciadv.aaw5873) merged independent alignments to two references containing the two known alleles to reduce reference bias. This is a more comparable approach to the one taken by the authors.

The speed of the alignment can be a limiting factor when dealing with large ancient DNA datasets. If the authors haven't done so in the supplementary materials, which I am unfortunately unable to access (wrong link in the reviewer pdf), then please provide some indication on run times compared to bwa.

With the parameters chosen, the authors find less accuracy for reference and alternative alignments compared to bwa, although the differences are small. They also see less total endogenous DNA with real data for vg. Why is that?

The population genetic analysis with D(vg, bwa, GRCh37, chimpanzee) should be D(vg, bwa, GRCh37, alternative), instead, to make the point you want to make. Deviations observed with the D including chimpanzee could otherwise also be explained by error differences between vg and bwa alignments. Note that D(vg, bwa, GRCh37, alternative) is nearly identical with the results from the previous section and this comparison can be shortened.

Minor/typos:

- Abstract: "sequencing reads are short, single-ended and frequently mutated". Reads do not mutate and can't be single-ended. This should be "molecules".
- Background, 2nd line: "past history" -> "history"
- Background: "Post-mortem damage occurs at a high rate, introducing mismatches in the tails of the short DNA molecules, which..." This sentence is confusing some facts. Deamination occurs throughout the

molecule, but it is more frequent at the molecule ends since it proceeds at a higher rate in single compared to double-stranded DNA and ancient molecules often have single-stranded overhangs at 5' or 3' end.

- Results, pg 3: "but essentially 1 in vg"; If the argument is that this difference from 1 is negligible, but the previous one is not, then a confidence interval would seem appropriate to show that. CI's may also be helpful for the alternative allele mapping fractions.

- Results, pg 5: Please decide for either "alternative allele mapped read fraction" or for the ratio of alternative to reference allele as a measure throughout the paper.

- Figure 4, lower panel: y-scale appears upside-down

Reviewer 2

Are you able to assess all statistics in the manuscript, including the appropriateness of statistical tests used? Yes, and I have assessed the statistics in my report.

Comments to author:

Review of Martiniano et al.

Removing reference bias in ancient DNA data analysis by mapping to a sequence variation graph

In this manuscript the authors evaluate the performance of their published variation graph software (vg) to the alignment of DNA sequences of ancient modern humans. Variation graphs, which allow known genetic variation to be included in the reference sequence, have previously been shown to improve sequence mapping and reduce reference bias compared to alignment to a linear reference. Since the sequence reads obtained from ancient samples are typically short, deaminated, and divergent from the reference genome, variant graphs should also improve the alignment of ancient DNA sequences. The authors explore this using simulated data as well as real sequence data from a set of ancient modern humans.

The authors have already previously published and evaluated their vg software for present-day modern human sequence alignment. There is no new software presented and therefore the only new aspect is a rather thin evaluation of the vg software on a different kind of data: ancient DNA sequences. Of course, this is an interesting domain and if the application of vg to sequence data from ancient modern humans shows substantial differences to performance on present-day human sequence data (not discussed at all), or if real analyses (split times, gene flow estimates etc) were much improved, this would be interesting to the community. However, I feel that the analyses presented currently fall short of demonstrating this clearly enough. I highlight major points which, if addressed, would strengthen the manuscript in the comments below.

General comments

The authors compare their vg software to bwa, which is commonly used for ancient DNA read alignment. To construct the variant graph they use genetic variation identified in the 1000 Genomes individuals. To align to the variant graph they use the mapper included in the vg software package. The resulting alignments are compared to alignments obtained using bwa with modified parameters. They evaluate primarily whether there is a reduction in the reference bias obtained by using the variant graph.

1. To date, the most substantial improvements in the alignment of ancient sequence reads have been approaches that simply increase the sequence divergence that is allowed. It is therefore important to show that the vg is an improvement over simply increasing the allowed sequence divergence. Some analyses that would provide information about this are present in the analysis, but this needs to be addressed in a more coherent and direct manner. For example: the use of different mapq is important: In using bwa, Hajdinjak et al. demonstrated that filtering bwa alignments for mapq ≥ 30 disproportionately removes

deaminated reads, but that reducing to $\text{mapq} \geq 25$ reduces this effect. For a comparison to state of the art it would therefore be preferable to test the use of $\text{mapq} \geq 25$, or even lower.

2. The authors demonstrate clearly that the major advantage of vg is in reducing bias at heterozygous sites. However, they also show that this improvement is only present for known sites that are actually included in the reference graph. They state: "bias in ancient DNA read alignment is mitigated at known sites by aligning against a variation graph" and „as expected, this difference was entirely due to mapping to previously identified 1000 Genomes Project sites present in the graph: new sites not in the graph showed no difference between the methods" This suggests that the reduction in reference bias applies only for a very limited set of samples: ancient modern humans that are inside the variation of present day samples used to construct the variation graph. There are certainly large numbers of genomes in this time-frame being generated, however, it would be good, given this constraint, for the authors to demonstrate more carefully that their statement that "for modern humans samples from the last 40,000 years this [that the graph depends on present-day variation] is not a major issue" What limitations are there? An assessment of the oldest set of Upper Paleolithic modern humans using the vg approach would be very informative here and may help determine what variation is most useful to include. For example, the high coverage genome of the ~45,000 year old Ust'Ishim individual would provide both an important comparison to the younger Yamnaya genome that is used for testing here - how much worse is it, given that it (and other earlier modern humans) is not a direct ancestor to present-day people? Also it may in fact be more suitable as a gold standard for defining a set of "correct HET genotypes" than the lower coverage Yamnaya individual used here.

3. I was surprised that the reduction in reference bias is also accompanied by "a slight decrease in accuracy in mapping the reference allele." In fact, the incorrect placement of reads carrying reference allele is quite a bit higher than is the case for bwa, and this effect is not only present at very short read lengths. Can the authors explain what is going on here? Is this also seen in the alignment of (longer) present-day human sequence data? Does this imply that in general reference graphs have an issue with off-target alignment, and that this is perhaps more severe the more variants are included? And finally, could this have an even greater effect on downstream analyses than reference bias?

4. Given this, I would have liked to have seen a more careful assessment of the false alignment rate of microbial sequence in the simulated dataset, and also in the real data. This could be tested by adding fragmented microbial genomes to the simulated reads and assessing whether false alignment of microbial reads (in different read-length bins) differs depending on either a fixed variation graph, or even modifying the amount of variation included in the graph.

5. The authors use a simple D-statistic to show that there is clearly less reference bias in the sequences aligned using vg. However, the authors do not compare the magnitude of the bias to that seen in the alignment of present-day human sequences (is the bias in ancient DNA sequences worse, or is it the same magnitude?) They also do not convincingly demonstrate that this bias (which is, in the end, small) has meaningful impact on downstream population genetic analyses, nor that there are analyses that would be sensitive to these differences. This is a critical point that needs to be addressed!

6. As the authors acknowledge, an approach to reducing reference bias in ancient DNA has already been published by Günther et al. The authors therefore motivate the need for their approach by making the point that the method of Gunther does not take into account indels. However, in their evaluation of the vg alignments, the authors do not convincingly show that the reduction in reference bias using vg is driven by differences in alignment around indels. This would seem an important analysis to demonstrate why the vg performs better - particularly why it performs better than simply increasing the number of mismatches allowed in alignment.

7. A point that is not clearly explained or discussed is how the vg can be so insensitive to deamination even at high rates (Figure 2). How exactly is deamination simulated here? If the plot were extended to higher deamination rates, where would the vg curve drop? It seems that even high rates of deamination have no effect on the vg alignment, which is surprising. Is this because the number of mismatches allowed is high? The authors should reassess alignment bias and off-target mapping including microbial sequences on reads with increasing amounts of deamination. This would demonstrate that the advantage

of vg does not come at the expense of e.g. false alignments of exogenous reads.

8. The authors state that "reads mapped with vg continue to show terminal deamination damage...although levels are slightly reduced." They then "attribute this reduction to differences in softclipping..." The authors should show evidence that this reduction is the result of soft-clipping (rather than a slight increase in modern human contamination which has also been shown to decrease the proportion of endogenous deamination in aligned reads).

Specific comments:

9. The authors should note more clearly somewhere in the main text that bwa is run using parameters modified to allow increased divergence and to turn off seeding.

10. The section that attempts to distinguish whether the improvement in reference bias is due to the graph or vg mapper needs to be explained more clearly for readers who are not necessarily familiar with the distinction being made. Providing a sentence that explains that the vg mapper can also map to a linear reference and then consistently referring to the aligner as the 'vg mapper' and not simply 'vg' would already help. Eg: "we also aligned simulated reads with THE VG MAPPER to the human linear reference genome GRCh37 ('vg linear') and compared the results obtained with vg MAPPER alignments to the 1000G graph."

11. Please provide a (supplementary) table including the sample identifiers and source publications for each of the real datasets. (Table 1 doesn't allow the reader to determine which publications the samples in later figures (eg: Figure 4) are from)

12. Pg 2. Line 11: "Post-mortem damage (PMD) of the DNA occurs at a high rate, introducing mismatches in the tails of the short DNA molecules, which are frequently in a single-stranded and relatively unprotected state." Technically damage does not introduce mismatches in the molecules. Either damage modifies the molecules by converting cytosines to uracils, or damage leads to mismatches in the alignment of a read to a reference.

13. Pg 3. Line 51: Please explain briefly for the reader what „the read modification protocol of Günther et al." is

14. Figure S8: Please provide a clear legend including labels explaining the colours

15. Figure S11: Please explain clearly in the legend why there are multiple points per sample

16. Figure 4: The authors explain the highly significant D-statistic for the final sample as being the result of low coverage. However, low coverage should lead to wider confidence intervals, not significant, positive D statistics.

17. The manuscript has a number of vague/unclear phrases. Please explain more clearly what is meant by statements such as "BWA makes more errors that are long range" Does this just mean bwa's errors are not clustered

18. Can the authors provide a brief comparison of the computational infrastructure required and the time required for graph construction/indexing and alignment using vg vs bwa

Reviewer 3

Are you able to assess all statistics in the manuscript, including the appropriateness of statistical tests used? Yes, and I have assessed the statistics in my report.

Comments to author:

It has been shown by many authors, among them also is the last author of the current manuscript, that when using a typical ultrafast aligner, such as the popular bwa (-mem or -aln), to map short reads against a single linear genome, the identification of the full extent of the variability of the sample's genome is biased towards the reference, ie. sensitivity as well as specificity is reduced.

This is of course also more than true for ancient DNA samples and the reconstruction of its respective underlying genome.

One of the problems of ancient DNA is that even for small genomes the reconstruction of the genome via de novo assembly is often not possible.

So far typically also bwa or other aligners for short reads are used together with a specific single, typically modern reference.

Sensitivity is a major issue for ancient DNA, since often very little DNA is available. If too many filtering steps need to be conducted prior to the actual mapping step, precious reads are lost and thus coverage is reduced to a level for which genotyping is either difficult or even impossible.

In this paper the authors suggest to use the software vg instead of bwa for example. The software vg basically uses the so-called variation graph rather than a single reference genome which represents the known genetic variation of the organism that is investigated and a vg-based mapper. They apply vg to the task of reconstructing human genomes from ancient DNA. They find that with vg the correct rate of calling alternate / non reference alleles is significantly increased. The authors show this for simulated as well as real ancient human genome data that have been generated with a variety of different experimental techniques. The decrease in bias for the reference allele has a number of consequences, for example for population genetic analyses that are typically conducted. The authors show that using vg also improves on these aspects as well.

In the paper several issues are addressed, in particular degree of deamination, for which the vg approach is most convincing in comparison to bwa-only. Thus the conclusions are adequately supported by the data shown.

The methods are mostly appropriate for the aim of the study, some minor adaptations are suggested below.

Also the methods section describes with enough detail to allow for replicating the analyses.

In summary, this is a nice paper, well-written, with only few comments that need to be address in a minor revision. I have listed them below.

And I wish to thank the authors to work with ancient DNA and the question of improved mapping-based assembly approaches. This work represents a significant advance over previously published studies and is of broad interest to others in the field.

Minor revision requests:

1a) Have the authors only tried bwa aln or also bwa mem? In my experience, bwa mem increases sensitivity also for ancient DNA samples.

1b) Along the lines of the mapping procedure (see also figure S1), left part. I noticed that samse for single-end reads has been used, not paired-end reads. Many ancient DNA projects use paired-end sequences, that are however, due to fragment lengths often much shorter than read lengths, first adapter trimmed and then the paired reads are "merged" if they have a negative insert size (see for example the pipeline EAGER, as described in Peltzer, A., Jäger, G., Herbig, A., Seitz, A., Kniep, C., Krause, J., & Nieselt, K. (2016). EAGER: efficient ancient genome reconstruction. *Genome biology*, 17(1), 60.) to increase the read length as well as improved base quality.

To my experience sensitivity and specificity is increased using this approach.

2) p. 4 lines 14-18: it seems that the % of incorrectly mapped reads is actually smaller for bwa than for vg? Why is this and how do the authors explain this? Is it because there is

3) p. 4 line 30: could you explain in more detail what are regions of reduced mappability? What is the reason for this?

4) One issue that the authors have referred to the impact of read length and mapping bias (see e.g. on p. 6, line 52). maybe you could add a reference to de Filippo et al. who have tried to find the optimal sweet spot between increasing the number of fragments for the analysis and excluding at the same time fragments which could align by chance and are therefore false positives:

de Filippo, C., Meyer, M., & Prüfer, K. (2018). Quantifying and reducing spurious alignments for the analysis of ultra-short ancient DNA sequences. *BMC biology*, 16(1), 121.

5) I suppose most differences when comparing the resulting genomes when being reconstructed using vg and/or bwa alone are seen in the SNPs themselves. I have not seen a table in the paper, whether the relative error rates are higher or similar for the three categories SNPs, indels and small complex variants. It would be nice to see this added.

6) The authors discussed the putative impact of the actual underlying VG graph for the reconstruction process.

In this case, the underlying variation graph was reconstructed from the phase 3 results of the 1000 Genomes project, which only comprise modern (healthy) humans. There are a number of ancient human genomes already reconstructed and available, so the question is how much is sensitivity as well as specificity even more increased when adding those to the vg. The authors argue that the current ancient human genomes have not shown any specific signals for ancient alleles, so currently no added value is gained.

Is this really true? The problem is that these genomes have been reconstructed with the typical linear single genome reference approach, so the reconstructed samples will have a bias towards modern reference alleles. If one could cumulatively add the ancient genomes based on the vg approach, thus with each added ancient human change the vg as being the union of 1000GP and those ancient human genomes, maybe this is not observed anymore.

Could the authors comment please?

7) Paleogenomics is obviously not restricted to the study of the human past.

I do not see why vg should only be applicable to human / eukaryote samples. What I wonder, how "deep" does the vg have to be to make a difference wrt to the mapping to a linear reference only? I suppose this is studied in the original vg paper, but maybe could be added here in the discussion when it comes to the point of applying vg to other ancient organisms.

Along these lines: In fact also a number of ancient genome projects are prokaryotic ones, where the bacteria of interest are mostly connected with severe epidemics, such as pest, leprosy or TB in the past. In addition to SNVs and/or indels, here also architectural changes are of interest, for which vg might not be the ideal data structure. Maybe the authors could add some sentences at the end of the discussion section that refers to alternative but similar approaches as vg, but need to be addressed when ancient bacterial genomes of an organism are to be reconstructed.

Minor typos:

sometimes the white space before a citation is missing, e.g. in p2. line 40, 48, 51, p3. 52,

p.8 line 52: snps -> SNPs

Dear Editor,

Thank you for your consideration and invitation to submit a revised manuscript. We have submitted a revised manuscript, including new text and analyses to address all the reviewers' comments. In particular, we test the additional standard alignment parameter settings and methods as requested by reviewers 1 and 2, and we show that our proposed approach using *vg* makes a major difference in the alignment of reads containing indels, affecting downstream analyses, demonstrating a significant advantage as requested by reviewer 2.

We include a description of changes in line with reviewer comments below:

Reviewer reports:

Reviewer #1: Martiniano et al. run the variance graph aligner *vg* on simulated and real ancient DNA datasets. As expected, the alignments show more equal representation of aligned reads supporting reference and (known) alternative alleles. Compared to one other method, that aims at alleviating reference bias in *bwa* alignments, they find an increased sensitivity.

While I have no doubt that *vg* is the right approach and can be useful for ancient DNA analyses, I have a number of issues with the methodology presented in the paper:

The authors use a mapping quality cutoff of 30 and parameters "-n 0.02 -l 1024" for *bwa*. More sensitive parameters (-n 0.01 instead of -n 0.02) have been used in the literature. It is also known that *bwa* sets mapping quality to 25 when reads align uniquely with the maximum edit distance. To reduce reference bias, a cutoff of MQ25 is often used. Fig. 2 shows quite clearly that the chosen *BWA* parameters perform poorly for both reference and alternative allele containing reads. It seems that more sensitive parameters, at least to the degree published in the literature, would be a fairer comparison.

The *bwa* alignment parameters we used initially (-n 0.02 and MQ30) were based on those recommended by Schubert et al., 2012, and were shown to increase sensitivity in read mapping in comparison with the default parameters. These have been widely used in high profile ancient DNA papers. Less stringent parameters for sequence alignment (-n 0.01 -o2) and mapping quality thresholds (MQ25) have also been used in the literature, in particular for very degraded and/or divergent sequence such as Neanderthal (Example: Peyregne et al., 2019). To address this request, we have added new results in the simulations section, including the percent of reads mapped after filtering with mapping quality >25 and using more sensitive parameters (-n 0.01 and -o 2), e.g. see Figure 2b and Supp.Fig.3. We confirm that using these more sensitive settings does increase the fraction of reads containing alternate alleles that map correctly, and consequently reduce mapping bias. We now discuss in the text how this mitigates against the bias in mapping to a linear reference, resulting in at least some cases to similar performance to the variation graph.

The greater sensitivity compared with Günther & Nettleblad (2019) is likely explained by the choice of *bwa* parameters (see above) and the fact that Günther & Nettleblad use a third allele when modifying the reference. In a different approach, Peyrégne et al. (2019) (DOI: 10.1126/sciadv.aaw5873) merged independent alignments to two references containing the two known alleles to reduce reference bias. This is a more comparable approach to the one taken by the authors.

We realigned our simulated data to the human reference genome and to a modified version of it, containing alternate SNPs at the Human Origins SNP sites, following the Peyrégne et al. (2019) workflow, and merging the resulting alignments. We have now included the results obtained with this workflow in our manuscript.

The speed of the alignment can be a limiting factor when dealing with large ancient DNA datasets. If the authors haven't done so in the supplementary materials, which I am unfortunately unable to access (wrong link in the reviewer pdf), then please provide some indication on run times compared to bwa.

Information about runtimes is present in the methods section of the main manuscript: "Read mapping with vg took approximately four (2.12-7.57) times longer than with bwa."

We added a table to the Supplementary Material (Table S4) which compares bwa and vg in terms of indexing and mapping of 10 million simulated reads and the following line to the Methods section: "We present a comparison of run times, memory and storage for indexing and read alignment between bwa and vg in Table S4 (and see [15] for additional estimates)."

With the parameters chosen, the authors find less accuracy for reference and alternative alignments compared to bwa, although the differences are small. They also see less total endogenous DNA with real data for vg. Why is that?

As seen in the original vg publication (Garrison et al, Nature Biotech, 2018), vg does lose a little sensitivity compared to bwa because some reads from repetitive regions that map to a unique place in the linear reference may map to more than one place in the graph reference. This becomes worse as more variants are introduced into the graph, which is why we only include variants with 1% frequency or more in our graph, as recommended by Garrison et al. We have added a couple of sentences about this into the paper. We also note that vg was developed to be comparable algorithmically to bwa mem, not bwa aln as used standardly for ancient DNA and in our paper. It is known that for short reads bwa mem is less accurate than bwa aln (Li 2013, bwa mem paper).

We added the following sentences to the discussion:

"An additional drawback of vg is the slightly lower sensitivity when compared to bwa in the mapping of reads in repetitive regions, as shown in [15]. When aligned to the linear reference, they map to a unique place in the linear reference, but in variation graphs they may map to more than one place. This becomes worse as more variants are introduced into the graph, which is why we included only variants with 1% frequency or more in our graph, as recommended by Garrison et al [15]."

The population genetic analysis with $D(\text{vg}, \text{bwa}, \text{GRCh37}, \text{chimpanzee})$ should be $D(\text{vg}, \text{bwa}, \text{GRCh37}, \text{alternative})$, instead, to make the point you want to make. Deviations observed with the D including chimpanzee could otherwise also be explained by error differences between vg and bwa alignments. Note that $D(\text{vg}, \text{bwa}, \text{GRCh37}, \text{alternative})$ is nearly identical with the results from the previous section and this comparison can be shortened.

We have calculated $D(\text{vg}, \text{bwa}, \text{GRCh37}, \text{alternative})$. We also obtain negative results for this statistic which corroborate a higher number of shared alleles between bwa and GRCh37 and between vg and the alternate allele, corroborating our previous findings of increased reference bias in bwa alignments. We added this analysis to the Supplementary Material.

Minor/typos:

- Abstract: "sequencing reads are short, single-ended and frequently mutated". Reads do not mutate and can't be single-ended. This should be "molecules".

This sentence now reads:

"However, the degraded nature of aDNA means that aDNA molecules are short and frequently mutated by post-mortem chemical modifications"

- Background, 2nd line: "past history" -> "history"

We made this change.

- Background: "Post-mortem damage occurs at a high rate, introducing mismatches in the tails of the short DNA molecules, which..." This sentence is confusing some facts. Deamination occurs throughout the molecule, but it is more frequent at the molecule ends since it proceeds at a higher rate in single compared to double-stranded DNA and ancient molecules often have single-stranded overhangs at 5' or 3' end.

We changed this to:

Post-mortem damage (PMD) of the DNA occurs at a high rate, introducing mismatches in DNA molecules, particularly in their tails which are frequently single-stranded or more exposed.

- Results, pg 3: "but essentially 1 in vg"; If the argument is that this difference from 1 is negligible, but the previous one is not, then a confidence interval would seem appropriate to show that. CI's may also be helpful for the alternative allele mapping fractions.

As recommended by the reviewer in the comment below, we have decided to show the fraction of alternate alleles in the final alignment as a measure throughout the paper rather than the ratio of alternative to reference allele. Therefore, we have changed that sentence. We also now present confidence intervals for alternate allele mapping fractions for both the simulations and the downsampling analysis.

- Results, pg 5: Please decide for either "alternative allele mapped read fraction" or for the ratio of alternative to reference allele as a measure throughout the paper.

We now present the alternate allele mapping fractions. This sentence now reads:

"However, because of reference bias, the fraction of alternate reads is on average 0.48267 (CI:0.48095-0.48438) in bwa aln (-n 0.02) but essentially 0.5 in vg 0.49987 (CI:0.49984-0.49991), supporting that vg alignment is not affected by reference bias in the same way as bwa."

- Figure 4, lower panel: y-scale appears upside-down

There is no lower panel to figure 4, just left and right. The values on the y-axis are categorical. The numerical scales are on the x-axis and run increasing from left to right as standard. Perhaps the reviewer rotated the page and was looking at this figure sideways on? We notice that in the originally submitted version the labels on the x-axis scales of the two panels are not consistently oriented. We have fixed that.

Reviewer #2: Review of Martiniano et al.

Removing reference bias in ancient DNA data analysis by mapping to a sequence variation graph

In this manuscript the authors evaluate the performance of their published variation graph software (vg) to the alignment of DNA sequences of ancient modern humans. Variation graphs, which allow known genetic variation to be included in the reference sequence, have previously been shown to improve sequence mapping and reduce reference bias compared to alignment to a linear reference. Since the sequence reads obtained from ancient samples are typically short, deaminated, and divergent from the reference genome, variant graphs should also improve the alignment of ancient DNA sequences. The authors explore this using simulated data as well as real sequence data from a set of ancient modern humans.

The authors have already previously published and evaluated their vg software for present-day modern human sequence alignment. There is no new software presented and therefore the only new aspect is a rather thin evaluation of the vg software on a different kind of data: ancient DNA sequences. Of course, this is an interesting domain and if the application of vg to sequence data from ancient modern humans shows substantial differences to performance on present-day human sequence data (not discussed at all), or if real analyses (split times, gene flow estimates etc) were much improved, this would be interesting to the community. However, I feel that the analyses presented currently fall short of demonstrating this clearly enough. I highlight major points which, if addressed, would strengthen the manuscript in the comments below.

General comments

The authors compare their vg software to bwa, which is commonly used for ancient DNA read alignment. To construct the variant graph they use genetic variation identified in the 1000 Genomes individuals. To align to the variant graph they use the mapper included in the vg software package. The resulting alignments are compared to alignments obtained using bwa with modified parameters. They evaluate primarily whether there is a reduction in the reference bias obtained by using the variant graph.

1. To date, the most substantial improvements in the alignment of ancient sequence reads have been approaches that simply increase the sequence divergence that is allowed. It is therefore important to show that the vg is an improvement over simply increasing the allowed sequence divergence. Some analyses that would provide information about this are present in the analysis, but this needs to be addressed in a more coherent and direct manner. For example: the use of different mapq is important: In using bwa, Hajdinjak et al. demonstrated that filtering bwa alignments for mapq ≥ 30 disproportionately removes deaminated reads, but that reducing to mapq ≥ 25 reduces this effect. For a comparison to state of the art it would therefore be preferable to test the use of mapq ≥ 25 , or even lower.

As discussed above for reviewer 1, we have added a figure to the main text (Fig 2b), and to the Supplementary Materials showing the percentage of reads aligned with bwa using different parameters and different mapping quality filter thresholds, including mapq>25. A table with error rates for the different parameters was also added to the Supplementary Materials.

2. The authors demonstrate clearly that the major advantage of vg is in reducing bias at heterozygous sites. However, they also show that this improvement is only present for known sites that are actually included in the reference graph. They state: "bias in ancient DNA read alignment is mitigated at known sites by aligning against a variation graph" and „as expected, this difference was entirely due to mapping to previously identified 1000 Genomes Project sites present in the graph: new sites not in the graph showed no difference between the methods" This suggests that the reduction in reference bias applies only for a very limited set of samples: ancient modern humans that are inside the variation of present day samples used to construct the variation graph. There are certainly large numbers of genomes in this time-frame being generated, however, it would be good, given this constraint, for the authors to demonstrate more carefully that their statement that "for modern humans samples from the last 40,000 years this [that the graph depends on present-day variation] is not a major issue" What limitations are there? An assessment of the oldest set of Upper Paleolithic modern humans using the vg approach would be very informative here and may help determine what variation is most useful to include. For example, the high coverage genome of the ~45,000 year old Ust'Ishim individual would provide both an important comparison to the younger Yamnaya genome that is used for testing here - how much worse is it, given that it (and other earlier modern humans) is not a direct ancestor to present-day people? Also it may in fact be more suitable as a gold standard for defining a set of "correct HET genotypes" than the lower coverage Yamnaya individual used here.

It is not correct that ancient modern humans that lie inside the variation of present day samples are a very limited subset of samples. Because the typical coalescent time in humans is approximately a million years, all modern humans yielding DNA (i.e. within the last 50,000 years) share the vast majority of their variation with present day humans. For example, 96.99% of high quality variants called de novo in Ust'Ishim chromosome 1 are found in the 1000 Genomes Project variant set. In fact this is less divergent than present-day Papuans from the 1000 Genomes Project data set, who have had more time to accumulate private mutations. The vast majority of ancient human DNA samples that are studied date from the last 15,000 years and are close to populations well represented in modern data sets.

3. I was surprised that the reduction in reference bias is also accompanied by "a slight decrease in accuracy in mapping the reference allele." In fact, the incorrect placement of reads carrying reference allele is quite a bit higher than is the case for bwa, and this effect is not only present at very short read lengths. Can the authors explain what is going on here? Is this also seen in the alignment of (longer) present-day human sequence data? Does this imply that in general reference graphs have an issue with off-target alignment, and that this is perhaps more severe the more variants are included? And finally, could this have an even greater effect on downstream analyses than reference bias?

This is not a consequence of using graph references compared to linear references. We believe it is a consequence of the vg alignment algorithm being based on bwa mem, which requires seed matches, rather than bwa aln which allows alignment without seeds, as is used in most ancient DNA studies and we used here. Reimplementation of the core alignment process is beyond the scope of this paper. We now discuss this more fully in the manuscript. We have run bwa mem to illustrate this effect. We still make the argument that the use of variation graphs can remove bias and is potentially valuable for ancient DNA mapping.

4. Given this, I would have liked to have seen a more careful assessment of the false alignment rate of microbial sequence in the simulated dataset, and also in the real data. This could be tested by adding fragmented microbial genomes to the simulated reads and assessing whether false alignment of microbial reads (in different read-length bins) differs depending on either a fixed variation graph, or even modifying the amount of variation included in the graph.

To address this issue, we have added analyses where we investigate the false alignment rate of different read mappers, using simulated reads from a set of microbial reference genomes identified in the Clovis sequence data (Rasmussen et al, 2014). We then mapped these reads with vg to the 1000GP graph and with bwa aln and bwa mem to the reference genome. We observe different patterns between the 3 aligners which we present in the results section.

5. The authors use a simple D-statistic to show that there is clearly less reference bias in the sequences aligned using vg. However, the authors do not compare the magnitude of the bias to that seen in the alignment of present-day human sequences (is the bias in ancient DNA sequences worse, or is it the same magnitude?) They also do not convincingly demonstrate that this bias (which is, in the end, small) has meaningful impact on downstream population genetic analyses, nor that there are analyses that would be sensitive to these differences. This is a critical point that needs to be addressed!

We have now included new analyses with indels which show a much more substantial magnitude in reference bias than SNP based analyses. First Figure S26 demonstrates the relative sensitivity to indels as a function of indel size. Second, we added a PCA analysis restricted to indels of the 1000 Genomes dataset, and estimate PCAs separately, restricting the analysis to alternate alleles identified in Yamnaya processed with bwa and Yamnaya with vg (Figure S27). We show that not only does vg recover a greater number of non-reference indels than bwa, it can also call longer variants. In fact, bwa did not call any indel larger than 6 base pairs, while vg does, and we show that this variation which is inaccessible to bwa, is ancestry informative. Analysis of indels is not just relevant for ancestry calculations, but also for study of the history of functional variants, some of which come in the form of indels, and we now point this out in the discussion.

6. As the authors acknowledge, an approach to reducing reference bias in ancient DNA has already been published by Günther et al. The authors therefore motivate the need for their approach by making the point that the method of Gunther does not take into account indels. However, in their evaluation of the vg alignments, the authors do not convincingly show that the reduction in reference bias using vg is driven by differences in alignment around indels. This would seem an important analysis to demonstrate why the vg performs better - particularly why it performs better than simply increasing the number of mismatches allowed in alignment..

As discussed above, we now present evidence that vg performs significantly better in the presence of indels. Figure 1 also shows an example where there is an indel. However we don't claim to demonstrate that the whole gain is from indels.

7. A point that is not clearly explained or discussed is how the vg can be so insensitive to deamination even at high rates (Figure 2). How exactly is deamination simulated here? If the plot were extended to higher deamination rates, where would the vg curve drop? It seems that even high rates of deamination have no effect on the vg alignment, which is surprising. Is this because the number of mismatches allowed is high? The authors should reassess alignment bias and off-target mapping including microbial sequences on reads with increasing amounts of deamination. This would demonstrate that the advantage of vg does not come at the expense of e.g. false alignments of exogenous reads.

Deamination was introduced with gargammel into sequencing reads based on deamination rates directly estimated from a set of 102 ancient DNA samples published by Allentoft et al. (2015).

We have now included analyses where we map simulated reads from microbial genomes, with deamination, and show that, at q50, spurious alignment of microbial reads (with read length greater or equal to 50 bp) is indeed slightly higher with vg, suggesting that the higher sensitivity of vg does come at the cost of slightly reduced specificity, but when at q60, vg's specificity is closer to bwa's. Importantly, bwa's specificity in the alignment of very short sequences (30-45 bp) is lower than vg's. We now present these results in the main text and added two supplementary figures (S8 and S9).

8. The authors state that "reads mapped with vg continue to show terminal deamination damage...although levels are slightly reduced." They then "attribute this reduction to differences in softclipping..." The authors should show evidence that this reduction is the result of soft-clipping (rather than a slight increase in modern human contamination which has also been shown to decrease the proportion of endogenous deamination in aligned reads.

We argue that the slightly lower frequencies of terminal deamination observed with vg when compared to bwa are due to more frequent read softclipping. In order to demonstrate this, we selected 50 bp chr11 reads from our simulated data which have a C nucleotide at the first or second base of the read. Next, we changed all first and second 'C' nucleotides to 'T' to mimic the effect of deamination. We aligned 50,000 of these artificially deaminated reads with bwa, vg and bwa mem and ran a mapDamage analysis. As expected, we observe an increase in softclipping in the first 2 bp of the 5'-end in the vg bam whereas with bwa, a C to T is instead observed. The mapDamage patterns obtained with bwa mem are identical to vg, and this is because vg was based on bwa mem's alignment algorithm.

Specific comments:

9. The authors should note more clearly somewhere in the main text that bwa is run using parameters modified to allow increased divergence and to turn off seeding.

We added this information to the methods section "... with parameters -l1024 (for disabling seed)..."

10. The section that attempts to distinguish whether the improvement in reference bias is due to the graph or vg mapper needs to be explained more clearly for readers who are not necessarily familiar with

the distinction being made. Providing a sentence that explains that the vg mapper can also map to a linear reference and then consistently referring to the aligner as the 'vg mapper' and not simply 'vg' would already help. Eg: "we also aligned simulated reads with THE VG MAPPER to the human linear reference genome GRCh37 ('vg linear') and compared the results obtained with vg MAPPER alignments to the 1000G graph."

When referring to vg linear, we added 'vg mapper' as recommended by the reviewer.

11. Please provide a (supplementary) table including the sample identifiers and source publications for each of the real datasets. (Table 1 doesn't allow the reader to determine which publications the samples in later figures (eg: Figure 4) are from)

We added this information to Table S2.

12. Pg 2. Line 11: "Post-mortem damage (PMD) of the DNA occurs at a high rate, introducing mismatches in the tails of the short DNA molecules, which are frequently in a single-stranded and relatively unprotected state." Technically damage does not introduce mismatches in the molecules. Either damage modifies the molecules by converting cytosines to uracils, or damage leads to mismatches in the alignment of a read to a reference.

This sentence now reads:

"Post-mortem damage (PMD) of the DNA occurs at a high rate, introducing mismatches in DNA molecules, particularly in their tails which are frequently single-stranded or more exposed."

13. Pg 3. Line 51: Please explain briefly for the reader what „the read modification protocol of Günther et al." is

We added the following sentence to the main text:

"We also applied the read modification protocol of Günther et al. to our bwa aln mapping data, in which reads overlapping a biallelic SNP are duplicated with the second copy carrying the other allele. If both reads map to the same region of the genome, then the mapping of the original, unmodified read is kept."

14. Figure S8: Please provide a clear legend including labels explaining the colours

We added the following information to the figure legend to the figure: 'Red: C to T substitutions; blue: G to A substitutions, and orange: soft-clipped bases.'

15. Figure S11: Please explain clearly in the legend why there are multiple points per sample

We added the following information to the figure legend to the Figure S14 to S18: 'We included 5 replicates per ancient sample to account for the randomness in pseudo-haploid genotype generation.'

16. Figure 4: The authors explain the highly significant D-statistic for the final sample as being the result of low coverage. However, low coverage should lead to wider confidence intervals, not significant, positive D statistics.

It is not significant in itself - the bars indicate standard deviations so this is under two standard deviations from zero. It is the systematic effect across samples that makes the results significant. We have clarified the text.

17. The manuscript has a number of vague/unclear phrases. Please explain more clearly what is meant by statements such as "BWA makes more errors that are long range" Does this just mean bwa's errors are not clustered.

We simplified the text in this location, removing “long range”, and have endeavoured to improve clarity elsewhere.

18. Can the authors provide a brief comparison of the computational infrastructure required and the time required for graph construction/indexing and alignment using vg vs bwa

We have added a brief summary of these numbers in the methods, and refer to new table S4 and to Garrison et al. who provide a table on compute times and memory required for each step of the graph construction, indexing and alignment.

Reviewer #3:

It has been shown by many authors, among them also is the last author of the current manuscript, that when using a typical ultrafast aligner, such as the popular bwa (-mem or -aln), to map short reads against a single linear genome, the identification of the full extent of the variability of the sample's genome is biased towards the reference, ie. sensitivity as well as specificity is reduced.

This is of course also more than true for ancient DNA samples and the reconstruction of its respective underlying genome.

One of the problems of ancient DNA is that even for small genomes the reconstruction of the genome via de novo assembly is often not possible.

So far typically also bwa or other aligners for short reads are used together with a specific single, typically modern reference.

Sensitivity is a major issue for ancient DNA, since often very little DNA is available. If too many filtering steps need to be conducted prior to the actual mapping step, precious reads are lost and thus coverage is reduced to a level for which genotyping is either difficult or even impossible.

In this paper the authors suggest to use the software vg instead of bwa for example. The software vg basically uses the so-called variation graph rather than a single reference genome which represents the known genetic variation of the organism that is investigated and a vg-based mapper. They apply vg to the task of reconstructing human genomes from ancient DNA. They find that with vg the correct rate of calling alternate / non reference alleles is significantly increased. The authors show this for simulated as well as real ancient human genome data that have been generated with a variety of different experimental techniques. The decrease in bias for the reference allele has a number of consequences, for example for population genetic analyses that are typically conducted. The authors show that using vg also improves on these aspects as well.

In the paper several issues are addressed, in particular degree of deamination, for which the vg approach is most convincing in comparison to bwa-only. Thus the conclusions are adequately supported by the data shown.

The methods are mostly appropriate for the aim of the study, some minor adaptations are suggested below.

Also the methods section describes with enough detail to allow for replicating the analyses.

In summary, this is a nice paper, well-written, with only few comments that need to be address in a minor revision. I have listed them below.

And I wish to thank the authors to work with ancient DNA and the question of improved mapping-based assembly approaches. This work represents a significant advance over previously published studies and is of broad interest to others in the field.

Minor revision requests:

1a) Have the authors only tried bwa aln or also bwa mem? In my experience, bwa mem increases sensitivity also for ancient DNA samples.

We now also look at bwa-mem. Sensitivity with bwa mem is higher than bwa aln -n 0.02, but so are error rates. Bwa mem's sensitivity is not higher anymore when comparing with bwa aln -n0.01 -o2. These results are now shown in Figure 2B.

1b) Along the lines of the mapping procedure (see also figure S1), left part. I noticed that samse for single-end reads has been used, not paired-end reads. Many ancient DNA projects use paired-end sequences, that are however, due to fragment lengths often much shorter than read lengths, first adapter trimmed and then the paired reads are "merged" if they have a negative insert size (see for example the pipeline EAGER, as described in Peltzer, A., Jäger, G., Herbig, A., Seitz, A., Kniep, C., Krause, J., & Nieselt, K. (2016). EAGER: efficient ancient genome reconstruction. *Genome Biology*, 17(1), 60.) to increase the read length as well as improved base quality.

To my experience sensitivity and specificity is increased using this approach.

Of the four datasets analysed in our manuscript, two were already trimmed by the authors of the study who provided the sequence data. Of the other two studies, one is paired-end (Damgaard 2018) and the other is single-end (Martiniano 2016). For the paired-end sequencing study (Damgaard 2018), we used AdapterRemoval, merging read pairs that overlap. This is essentially the procedure that is implemented in EAGER, and therefore, a comparison with that workflow is not necessary.

2) p. 4 lines 14-18: it seems that the % of incorrectly mapped reads is actually smaller for bwa than for vg? Why is this and how do the authors explain this? Is it because there is

Addressed above in point 3 of reviewer 2.

3) p. 4 line 30: could you explain in more detail what are regions of reduced mappability? What is the reason for this?

Regions of reduced mappability are regions of the genome with mappability below 1, according to the Mappability 50 track obtained from the UCSC table browser. A mappability of 0.5 means that a 50 bp read can be mapped equally well to two different places in the human reference genome. We added a sentence to explain this in the figure legend in the Supplementary Material.

4) One issue that the authors have referred to the impact of read length and mapping bias (see e.g. on p. 6, line 52). maybe you could add a reference to de Filippo et al. who have tried to find the optimal sweet spot between increasing the number of fragments for the analysis and excluding at the same time fragments which could align by chance and are therefore false positives:

We now cite the de Filippo et al. (2018) paper when discussing our new results about the relationship of read length and spurious microbial contaminant read alignment.

5) I suppose most differences when comparing the resulting genomes when being reconstructed using vg and/or bwa alone are seen in the SNPs themselves. I have not seen a table in the paper, whether the relative error rates are higher or similar for the three categories SNPs, indels and small complex variants. It would be nice to see this added.

We now provide a figure S26 which shows relative performance on indels.

6) The authors discussed the putative impact of the actual underlying VG graph for the reconstruction process.

In this case, the underlying variation graph was reconstructed from the phase 3 results of the 1000 Genomes project, which only comprise modern (healthy) humans. There are a number of ancient human

genomes already reconstructed and available, so the question is how much is sensitivity as well as specificity even more increased when adding those to the vg. The authors argue that the current ancient human genomes have not shown any specific signals for ancient alleles, so currently no added value is gained.

There seems to have been a misunderstanding. We said that most variation from the linear reference that is seen in ancient samples is also present in the 1000 Genomes Project data set. This is a consequence of the fact that modern human samples, unlike archaics, have shared with extant modern humans most of the history since the most recent common ancestor of all modern humans, which is typically around a million years ago ($0.5 * \text{heterozygosity} * \text{generation_time} / \text{mutation_rate} = 0.5 * 0.001 * 30 / 1.25e-8 = 0.015 / 1.25e-8 = 1.2e6$ years).

Is this really true? The problem is that these genomes have been reconstructed with the typical linear single genome reference approach, so the reconstructed samples will have a bias towards modern reference alleles. If one could cumulatively add the ancient genomes based on the vg approach, thus with each added ancient human change the vg as being the union of 1000GP and those ancient human genomes, maybe this is not observed anymore.

Could the authors comment please?

We address this in the discussion in the context of archaic genomes, and also mention adding additional ancient material to the graph.

7) Paleogenomics is obviously not restricted to the study of the human past.

I do not see why vg should only be applicable to human / eukaryote samples. What I wonder, how "deep" does the vg have to be to make a difference wrt to the mapping to a linear reference only? I suppose this is studied in the original vg paper, but maybe could be added here in the discussion when it comes to the point of applying vg to other ancient organisms.

Along these lines: In fact also a number of ancient genome projects are prokaryotic ones, where the bacteria of interest are mostly connected with severe epidemics, such as pest, leprosy or TB in the past. In addition to SNVs and/or indels, here also architectural changes are of interest, for which vg might not be the ideal data structure. Maybe the authors could add some sentences at the end of the discussion section that refers to alternative but similar approaches as vg, but need to be addressed when ancient bacterial genomes of an organism are to be reconstructed.

In fact the vg structure can represent and map to arbitrary structural variation, as demonstrated in the recent paper by Hickey et al. (PMID 32051000). We have added a short paragraph to the discussion concerning applications to other species and structural variation..

Minor typos:

sometimes the white space before a citation is missing, e.g. in p2. line 40, 48, 51, p3. 52,

p.8 line 52: snps -> SNPs

We have corrected "snps" to "SNPs" and added whitespace before citations as suggested.

Additional changes:

We added Torsten Günther and Gabriel Renaud to the Acknowledgments section.

Second round of review

Reviewer 2

The authors have revised their manuscript to include a number of the new analyses suggested by reviewers, as well as to show that the detection of indels in ancient DNA data is a particularly compelling use case for vg.

To me there are two aspects that still need some work in order to make the contributions of variant graph alignments in ancient DNA clearer:

1. The main conclusions of the manuscript.

The new analysis of vg alignments in CCR5 adds to the existing evidence that the most compelling advantage of vg for aDNA is indel detection. Further, the newly added analysis of the BWA alignment to the linear reference using more sensitive parameters shows that the improvement in SNP reference bias offered by vg is not as substantial as reported in the initial manuscript. I would therefore urge the authors to revise the manuscript to focus more on indel detection rather than reference bias reduction at SNPs.

2. The consistency of analyses performed

Multiple measures are generally taken into account when evaluating alignment approaches for aDNA. These include reference bias, mapping accuracy, the spurious alignment of microbial reads and the speed/memory required. This manuscript is focussed on the removal of reference bias. However, in optimizing this parameter, it is critical to assess the effects on the other measures. In this case, particularly interesting are the effects on (i) mapping accuracy and (ii) the spurious alignment of microbial reads.

For readers to really be able to evaluate the results fairly and to see the trade-offs, it would be helpful if the authors provide analyses of each measure (reference bias, mapping accuracy, microbial alignment rate and speed/memory) on sets of data analyzed in a consistent set of ways. For instance, there are four different conditions for the alignments in figure 2a, 12 in figure 2b, eight in figure 3, and more different numbers of parameters/conditions across the supplementary material, tables and figures. It would also be useful if the authors make some recommendation on mapq filtering for applying the vg to ancient DNA

Table S1 provides a very nice comparison of the different alignment approaches on read mapping accuracy. It is clear from this that mapping to a linear reference with bwa, and specifically with `bwa aln -n 0.01 -o 2 + mapq25` is actually more accurate than vg at the recommended filtering of `mapq50`. A similar table evaluating the same alignment approaches with respect to their effect on the spurious alignment of microbial reads over all length bins (no need to deaminate them!) would be a very valuable addition to the paper. As far as I can see, the evaluation of microbial misalignment in Figure S8 only includes the bwa parameter set that performs very poorly for reducing reference bias (`bwa aln -n 0.02 + mapq30`). It is therefore still difficult to determine whether more microbial reads are aligned when using the optimal parameters for vg

In response to the reviewers' questions about reference bias the authors now include a comparison of vg to `bwa aln` with multiple different parameters sets that include the two different parameters settings (below), and test the effect of a less stringent mapq filtering

- * `bwa aln -n 0.02`
- * `bwa aln -n 0.01 -o 2`

In Figure S3 it is clear that using `bwa aln -n 0.01 -o 2 + mapq25 filter` - as has been done in many aDNA studies - is comparable in terms of the % of mapped reads to using vg graph `mapq50`. There is therefore

no major gain offered by vg for reference bias at heterozygous SNPs when compared to a widely used approach.

In contrast, `bwa aln -n 0.02+mapq30` - the parameters set that the authors compare throughout the paper - is clearly highly biased. This is also clear when looking at Figure 2B where `bwa aln -n 0.02+mapq30` is clearly the worst performing parameter set for bwa. I am therefore puzzled that the authors choose to compare bwa using these parameters for most of the evaluations of bwa/linear reference presented in the main text and present it in Figure 2A.

I recommend strongly that the authors either carry both `bwa aln -n 0.02+mapq30` and `bwa aln -n 0.01 -o 2 + mapq25` through all analyses, or that they replace `bwa aln -n 0.02+mapq30` with the more sensitive `bwa aln -n 0.01 -o 2 + mapq25` in all analyses. It seems unfair to compare only the least good bwa parameter set to the best vg set.

Reviewer 3

I have not been as critical as my colleague reviewers, as I do see the potential of vg mapping approaches in principle also for ancient DNA samples.

The authors have made a great effort to reply to all issues raised by the 3 reviewers.

After first reading the revised manuscript I had wondered whether the authors could actually pinpoint a mutation / variant which has been shown of being of clinical importance and which can be found in ancient samples using vg but not bwa. I have then been informed by the editor that the authors have indeed addressed this question and not just wondered but also found such an example. This is a definite plus for the new method.

The authors have specified now the runtime of vg in comparison to the other methods. It needs longer, and obviously also the indexing needs much more time. Since the variant graph should actually be “constantly” updated (with every variant with a minimal MAF), this is an issue, but I do not consider it a big “problem”.

Nonetheless, runtimes are only mentioned in the method section and never taken up for example in the discussion.

As for my requests in my first review, the only question (does not need to be addressed in a revised paper) is the following: as explained by the authors under the hood of vg the alignment method is based on bwa-mem. Of course now seeing new methods such as minimap etc. being commonly applied, but also `bwa-aln` for very short reads from ancient samples, I wonder whether the authors see a possibility to offer more flexibility wrt to the chosen aligner during vg alignment.

A small observation in Figure S11: do you know why there is a small bump in increased damage frequency at pos. +6 and -6 when using vg (independent of mapq).

Thus, the paper needs only some more editorial revisions.

Minor editorial requests to be revised:

Generally the authors should go over the manuscript and make sure that consistently `bwa aln` and `bwa mem` are mentioned. An example is the Method section “Datasets and sequence data processing” and many more places.

Abstract: “which includes all the alternative variants at each genetic locus.” -> all alternatives variants is not quite precise, it should be made clear that the variant graph can only represent currently known variants.

p.2 line 54: “Here we apply vg and bwa aln” -> bwa (mem and aln)

p.3 line 53: however, as expected, decreasing mapping quality results an increase in error rates
-> in an increase ...

p. 3 line: “We then mapped simulated sequences back to the” -> simulated reads

p.3 line 42-48: “However, because of reference bias, the fraction of alternate reads is on average 0.48267 (CI:0.48095-0.48438) in bwa aln (-n 0.02) but essentially 0.5 in vg 0.49987 (CI:0.49984-0.49991), supporting that vg alignment is not affected by reference bias in the same way as bwa. When relaxing the parameters in bwa aln (-n 0.01 -o 2), we observe a better representation of alternate alleles 0.49702 (CI:0.49657-0.49747) in the final alignment, but the bias towards the reference is still slightly higher than with vg (Figure 2b).”

This does not read perfectly, please fix the sentences with the numbers of the alleles ratio, most of them should be in brackets, preceding the confidence interval numbers, or reformulate.

p.4 line 21ff: “We then processed our simulated data with a different workflow for removing reference bias [15],...” I suggest to reformulate as follows:

We then processed our simulated data with a different workflow for removing reference bias as suggested by Peyregne et al. [15]: reads are mapped with bwa aln to two versions of the human reference genome ...
”

p.4 line 33-38: a couple of “respectively” are missing.

p.4 line 49-52: I suggest to use 1.2e-04 % etc. rather than 0.000012 % for better readability and comparability.

p.5 line 23-46. these two new paragraphs can be slightly rewritten, such that in the first paragraph only the issue about length of reads and falls mapping rate is presented and discussed, while the second shorten paragraph should only address the question about level of deamination (independence in fact). Lines 43-55 are fine.

p.8 line 28: (Peyregne et al 2019) -> change to [15] (i.e. \cite{ }) also change Peyregne to Peyrégne everywhere

p. 8 line 39-40: “One complication of our analysis ...are not directly comparable between vg and bwa aln.” -> between vg and bwa aln or bwa-mem.

Further minor typo:

freebayes -> FreeBayes (well fair enough, the authors of the method are not consistent themselves, while they call it “FreeBayes” in their original arXiv paper, on their GitHub page it is freebayes. The authors may choose to leave it as it is.

Following the reviewers' comments, we have now revised our analyses to include the best performing bwa aln parameters -n 0.01 -o 2 in all the main analyses for comparison with vg.

Figure 2, where we show results with simulated reads, was updated to include the '-n 0.01 -o 2' parameters (Fig 2b), and multiple variations of mapping quality filters and post-processing workflows to address reference bias. In addition, we added a panel Fig. 2d and a table (Table S1) containing the alternate allele fractions, and the percentage of mapped reads by the different software using various parameters in Fig2c.

We have edited the manuscript to move the overall emphasis towards vg's superior ability for mapping reads containing indel variation, and we propose to change the title of the manuscript to "Removing reference bias and improving indel calling in ancient DNA data analysis by mapping to a sequence variation graph", in order to reflect the superior performance of vg for recovering indels.

In addition, in Figure 3 (downsampling experiment), we added the Gunther and Nettleblad read modification method applied to the 'bwa aln -n 0.01 -o 2' parameters for a more complete assessment. We also show a comparison between bwa aln and vg in terms of indel detection in Fig 3b.

We made changes to our previous Figure 4 (D-statistics) and moved it to the supplementary material, where we now show a comparison between vg and bwa aln -n 0.02 and bwa aln -n 0.01 -o 2, as requested by reviewer number 2. With 'bwa aln -n 0.02', we observe a bias towards the reference, but this bias is now greatly attenuated when using the more relaxed bwa parameters '-n 0.01 -o 2'. We have also added a few sentences in the manuscript about these findings.

We have relabelled many figures and made changes to the figure legends to clarify which bwa parameters were used for each analysis.

Detailed comments with respect to reviewer reports:

Reviewer #2: The authors have revised their manuscript to include a number of the new analyses suggested by reviewers, as well as to show that the detection of indels in ancient DNA data is a particularly compelling use case for vg.

To me there are two aspects that still need some work in order to make the contributions of variant graph alignments in ancient DNA clearer:

1. The main conclusions of the manuscript.

The new analysis of vg alignments in CCR5 adds to the existing evidence that the most compelling advantage of vg for aDNA is indel detection. Further, the newly added analysis of the BWA alignment to the linear reference using more sensitive parameters shows that the improvement in SNP reference bias offered by vg is not as substantial as reported in the

initial manuscript. I would therefore urge the authors to revise the manuscript to focus more on indel detection rather than reference bias reduction at SNPs.

We added Figure 3b, where we show results of the downsampling experiment focused on indels.

We have rewritten our manuscript in several places to reduce our claims of substantial advantage of vg for SNPs comparatively to bwa aln -n0.01 -o2, and highlight vg's clear advantage in mapping reads containing indels.

2. The consistency of analyses performed

Multiple measures are generally taken into account when evaluating alignment approaches for aDNA. These include reference bias, mapping accuracy, the spurious alignment of microbial reads and the speed/memory required. This manuscript is focussed on the removal of reference bias. However, in optimizing this parameter, it is critical to assess the effects on the other measures. In this case, particularly interesting are the effects on (i) mapping accuracy and (ii) the spurious alignment of microbial reads.

For readers to really be able to evaluate the results fairly and to see the trade-offs, it would be helpful if the authors provide analyses of each measure (reference bias, mapping accuracy, microbial alignment rate and speed/memory) on sets of data analyzed in a consistent set of ways. For instance, there are four different conditions for the alignments in figure 2a, 12 in figure 2b, eight in figure 3, and more different numbers of parameters/conditions across the supplementary material, tables and figures. It would also be useful if the authors make some recommendation on mapq filtering for applying the vg to ancient DNA

For our simulated data, we can run a wide range of parameters relatively quickly, and therefore we test bwa aln, bwa mem, vg linear and vg graph. We now show a complete comparative assessment of these in terms of percentage of reads aligned, alternate allele fraction and error rates in Fig2 and Table S1, and throughout the supplementary material.

We now include in Figure 2 and 3 and in the supplementary material comparisons between vg graph, vg linear, bwa mem, bwa aln -n 0.01 -o 2 and bwa aln -n 0.02. Where appropriate, we show different mapping quality filters. In some cases, exploring such a wide array of parameters is not necessary, and we limit our comparisons to the most widely used bwa aln parameters of bwa aln -n 0.01 -o 2 and bwa aln -n 0.02.

We added the following recommendation: For vg in particular, we recommend imposing a minimum mapping quality filter of 50 for obtaining comparable to those of bwa aln (albeit slightly higher), comparable sensitivity and to minimise the spurious alignment of microbial reads"

Table S1 provides a very nice comparison of the different alignment approaches on read mapping accuracy. It is clear from this that mapping to a linear reference with bwa, and specifically with `bwa aln -n 0.01 -o 2 + mapq25` is actually more accurate than vg at the recommended filtering of mapq50.

A similar table evaluating the same alignment approaches with respect to their effect on the spurious alignment of microbial reads over all length bins (no need to deaminate them!) would be a very valuable addition to the paper. As far as I can see, the evaluation of microbial misalignment in Figure S8 only includes the bwa parameter set that performs very poorly for reducing reference bias (`bwa aln -n 0.02 + mapq30`). It is therefore still difficult to determine whether more microbial reads are aligned when using the optimal parameters for vg

We added table S3 which shows the results of microbial read misalignment to the human reference genome over all length bins with the updated parameters `bwa aln -n 0.01 -o 2`. We have updated our analyses shown in Figure S8 and Figure S9 with the parameters '`bwa aln -n 0.01 -o 2`'.

We added the following sentence to the main text:

"Relaxing bwa aln parameters to '`-n 0.01 -o 2`' causes an increase (2.372%, `mapQ >= 25`) in the percentage of incorrectly mapped microbial reads compared to '`-n 0.02`' (Table S3, Figure S9)"

In response to the reviewers' questions about reference bias the authors now include a comparison of vg to bwa aln with multiple different parameters sets that include the two different parameters settings (below), and test the effect of a less stringent mapq filtering

- `bwa aln -n 0.02`
- `bwa aln -n 0.01 -o 2`

In Figure S3 it is clear that using `bwa aln -n 0.01 -o 2 + mapq25` filter – as has been done in many aDNA studies - is comparable in terms of the % of mapped reads to using vg graph mapq50. There is therefore no major gain offered by vg for reference bias at heterozygous SNPs when compared to a widely used approach.

In contrast, `bwa aln -n 0.02+mapq30` - the parameters set that the authors compare throughout the paper – is clearly highly biased. This is also clear when looking at Figure 2B where `bwa aln -n 0.02+mapq30` is clearly the worst performing parameter set for bwa. I am therefore puzzled that the authors choose to compare bwa using these parameters for most of the evaluations of bwa/linear reference presented in the main text and present it in Figure 2A.

I recommend strongly that the authors either carry both `bwa aln -n 0.02+mapq30` and `bwa aln -n 0.01 -o 2 + mapq25` through all analyses, or that they replace `bwa aln -n 0.02+mapq30` with the more sensitive `bwa aln -n 0.01 -o 2 + mapq25` in all analyses. It seems unfair to compare only the least good bwa parameter set to the best vg set.

The settings of `bwa -n 0.02` and disabling seed are recommended in Schubert et al 2012, and this is why we chose to compare these with vg. We acknowledge that many papers have now started using more relaxed parameters (`bwa aln -n 0.01 -o 2 + mapq25`), and we

now update our analyses, where relevant, to show both `bwa aln -n 0.02+mapq30` and `bwa aln -n 0.01 -o 2 + mapq25`.

Reviewer #3: I have not been as critical as my colleague reviewers, as I do see the potential of vg mapping approaches in principle also for ancient DNA samples.

The authors have made a great effort to reply to all issues raised by the 3 reviewers.

After first reading the revised manuscript I had wondered whether the authors could actually pinpoint a mutation / variant which has been shown of being of clinical importance and which can be found in ancient samples using vg but not bwa. I have then been informed by the editor that the authors have indeed addressed this question and not just wondered but also found such an example. This is a definite plus for the new method.

The authors have specified now the runtime of vg in comparison to the other methods. It needs longer, and obviously also the indexing needs much more time. Since the variant graph should actually be “constantly” updated (with every variant with a minimal MAF), this is an issue, but I do not consider it a big “problem”.

Nonetheless, runtimes are only mentioned in the method section and never taken up for example in the discussion.

We now mention runtimes in the discussion

“Furthermore, read alignment with vg takes ~4.6x longer than with `bwa aln -n 0.01 -o 2` and 27.7x than `bwa mem`, and the memory requirements for both indexing the graph and read mapping can be much more substantial than for indexing a linear reference genome, depending on the number of variants included.”

As for my requests in my first review, the only question (does not need to be addressed in a revised paper) is the following: as explained by the authors under the hood of vg the alignment method is based on `bwa-mem`. Of course now seeing new methods such as `minimap` etc. being commonly applied, but also `bwa-aln` for very short reads from ancient samples, I wonder whether the authors see a possibility to offer more flexibility wrt to the chosen aligner during vg alignment.

The alignment approach in `vg map` was inspired by `bwa mem` in that it uses similar seeding, alignment, and scoring approaches. This was done to ease comparison between the methods during the development of `vg map`, and also reflects our confidence in the quality of `bwa mem`.

The vg toolkit includes several other read mappers, including the "multipath mapper" `vg mpmap` and an exact-match based method, `vg gaffe`. The multipath mapper can describe the mapping of a sequence against a genome graph using a directed, acyclic set of paths through the graph. This allows it to represent local alignment uncertainty with respect to variation, which can be useful when alignment is guided by haplotype matching in a haplotype prior encoded in an associated GBWT (graph Burrows-Wheeler transform). `vg mpmap` has similar performance characteristics to `vg map`, with added complexity which can be useful for matching long haplotypes but is unlikely to be helpful for short aDNA reads. In contrast, `vg gaffe` is an experimental method designed to be very fast for short, high-quality

Illumina reads. It also uses the GBWT to guide alignment, but in contrast to map and mpmc it depends on a minimizer index. The minimizer index is likely to reduce its sensitivity for aDNA reads with respect to standard map as used in our manuscript

Several other variation graph mappers exist. The most prominent is GraphAligner. This method produces alignments in the same native formats used by vg. However, it is designed for long reads, and its seeding method (which defaults to a minimizer index) is unlikely to work well for these short reads.

We expect that the number of alignment methods that work on variation graphs will increase in the future, and hope that our work encourages the development of approaches specifically tuned to the needs of ancient DNA analysis. However we have used the one which is most mature and we believe most appropriate for aDNA mapping in our manuscript.

A small observation in Figure S11: do you know why there is a small bump in increased damage frequency at pos. +6 and -6 when using vg (independent of mapg).

We are not certain of the cause, but we believe this reflects differences in the soft clipping behavior relative to bwa mem and bwa aln. In vg map, we use a score bonus for reads that map full-length, while in bwa a penalty is applied to reads that don't map full length. The effects of both should be similar, but the implementations are different and may give rise differences like the one observed here. The specific bonus used in vg map (5) is curiously similar to the number of matching bases at the end of the read before the damage increase. We assume that reads with damage closer to their ends are relatively more likely to be soft clipped, reducing the apparent damage rate in the regions from pos +5 and -5 to the ends of the reads. Investigating this further is beyond the scope of this paper.

Thus, the paper needs only some more editorial revisions.

Minor editorial requests to be revised:

Generally the authors should go over the manuscript and make sure that consistently bwa aln and bwa mem are mentioned. An example is the Method section "Datasets and sequence data processing" and many more places.

We have changed the text to address this issue.

Abstract: "which includes all the alternative variants at each genetic locus." -> all alternative variants is not quite precise, it should be made clear that the variant graph can only represent currently known variants.

We have changed the text as follows to address this issue: "variation graph which includes known alternative variants at each genetic locus"

p.2 line 54: "Here we apply vg and bwa aln" -> bwa (mem and aln)

We do not use bwa mem for processing real data, only for simulations. Therefore we did not change this instance. We did follow the reviewer's suggestion in another place in this text: "First we used simulation to examine the impact of post-mortem deamination (PMD) in vg and bwa (aln mem) read alignment"

p.3 line 53: however, as expected, decreasing mapping quality results an increase in error rates

-> in an increase ...

We have changed the text as suggested.

p. 3 line: "We then mapped simulated sequences back to the" -> simulated reads

We have changed the text as suggested.

p.3 line 42-48: "However, because of reference bias, the fraction of alternate reads is on average 0.48267 (CI:0.48095-0.48438) in bwa aln (-n 0.02) but essentially 0.5 in vg 0.49987 (CI:0.49984-0.49991), supporting that vg alignment is not affected by reference bias in the same way as bwa. When relaxing the parameters in bwa aln (-n 0.01 -o 2), we observe a better representation of alternate alleles 0.49702 (CI:0.49657-0.49747) in the final alignment, but the bias towards the reference is still slightly higher than with vg (Figure 2b)."

This does not read perfectly, please fix the sentences with the numbers of the alleles ratio, most of them should be in brackets, preceding the confidence interval numbers, or reformulate.

We added brackets to confidence intervals.

p.4 line 21ff: "We then processed our simulated data with a different workflow for removing reference bias [15],..." I suggest to reformulate as follows:

We then processed our simulated data with a different workflow for removing reference bias as suggested by Peyregne et al. [15]: reads are mapped with bwa aln to two versions of the human reference genome ...

We have changed the text as suggested.

p.4 line 33-38: a couple of "respectively" are missing.

We are sorry, we have read the text again and can not see where the reviewer wants us to add them. We would be happy to look again with more context and add in proof if necessary.

p.4 line 49-52: I suggest to use 1.2e-04 % etc. rather than 0.000012 % for better readability and comparability.

We changed this to 1.2 per million etc.

p.5 line 23-46. these two new paragraphs can be slightly rewritten, such that in the first paragraph only the issue about length of reads and falls mapping rate is presented and discussed, while the second shorten paragraph should only address the question about level of deamination (independence in fact). Lines 43-55 are fine.

We have rewritten these paragraphs to address this issue.

p.8 line 28: (Peyregne et al 2019) -> change to [15] (i.e. `\cite{}`) also change Peyregne to Peyrégne everywhere

We have made these changes.

p. 8 line 39-40: “One complication of our analysis ...are not directly comparable between vg and bwa aln.” -> between vg and bwa aln or bwa-mem.

We did not change this sentence because it is correct in its original form. Mapping qualities are not directly comparable between vg and bwa aln, but they are between vg and bwa-mem.

Further minor typo:

freebayes -> FreeBayes (well fair enough, the authors of the method are not consistent themselves, while they call it “FreeBayes” in their original arXiv paper, on their GitHub page it is freebayes. The authors may choose to leave it as it is.

We changed freebayes to FreeBayes.